# Cas phosphorylation regulates focal adhesion assembly

Saurav Kumar, Amanda Stainer, Julien Dubrulle, Christopher Simpkins†, Jonathan A Cooper*

Fred Hutchinson Cancer Center, Seattle, United States

**Abstract** Integrin-mediated cell attachment rapidly induces tyrosine kinase signaling. Despite years of research, the role of this signaling in integrin activation and focal adhesion assembly is unclear. We provide evidence that the Src-family kinase (SFK) substrate Cas (Crk-associated substrate, p130Cas, BCAR1) is phosphorylated and associated with its Crk/CrkL effectors in clusters that are precursors of focal adhesions. The initial phospho-Cas clusters contain integrin β1 in its inactive, bent closed, conformation. Later, phospho-Cas and total Cas levels decrease as integrin β1 is activated and core focal adhesion proteins including vinculin, talin, kindlin, and paxillin are recruited. Cas is required for cell spreading and focal adhesion assembly in epithelial and fibroblast cells on collagen and fibronectin. Cas cluster formation requires Cas, Crk/CrkL, SFKs, and Rac1 but not vinculin. Rac1 provides positive feedback onto Cas through reactive oxygen, opposed by negative feedback from the ubiquitin proteasome system. The results suggest a two-step model for focal adhesion assembly in which clusters of phospho-Cas, effectors and inactive integrin β1 grow through positive feedback prior to integrin activation and recruitment of core focal adhesion proteins.

*For correspondence:
jcooper@fhcrc.org

Present address: †BCMB Program, Johns Hopkins University School of Medicine, Baltimore, United States

## Editor's evaluation

This important study advances our understanding of adhesion formation in migrating cells by showing that clustering of the adaptor protein Cas and its binding partners represents the initial step in adhesion formation that occurs before integrin clustering. The evidence supporting the conclusions is convincing overall, although in a few cases, quantifications are based on limited datasets.

## Introduction

Cell migration on extracellular matrix (ECM) involves the repeated assembly and disassembly of integrin-mediated cell–ECM adhesions (*Hynes, 2002*). Integrins are heterodimers of α and β chains that can switch between inactive and active conformations. Integrin activation exposes binding sites for ECM outside the cell and for specific integrin-tail-binding proteins inside the cell. The latter can associate with other proteins to form focal adhesions that link integrins to the actin cytoskeleton, providing traction forces for actomyosin-driven cell movement (*Iwamoto and Calderwood, 2015*; *Kanchanawong and Calderwood, 2022*, *Moser et al., 2009*). At the ultrastructural level, focal adhesions are heterogeneous, with nanoclusters of active integrins and integrin tail-associated proteins closest to the membrane, F-actin and actin-associated proteins on top, and mechanosensing force transducers sandwiched between (*Kanchanawong et al., 2010*; *Legerstee and Houtsmuller, 2021*). Forces generated by actin polymerization and actomyosin contractility partially unfold the force transducers, exposing binding sites for other focal adhesion proteins and stabilizing the structure (*Wolfenson et al., 2019*). For example, talin is a conformationally sensitive protein that binds integrin through its head domain and actin through sites in its tail. Tension

between the talin head and tail regions exposes binding sites for vinculin and other structural and regulatory proteins (*Bachmann et al., 2023*). In turn, binding to talin exposes actin-binding sites on vinculin, building links between integrins and actin filaments, providing resistance to contractile forces and anchorage for cell movement. Even though these mechanical principles are well understood, it is unclear whether they explain the early stages of adhesion assembly, when integrin clusters may be too small to develop sufficient force and where inside-out signaling, membrane lipid microdomains, the glycocalyx, and actin polymerization may also be important (*Coyer et al., 2012*; *Henning Stumpf et al., 2020*).

In addition to binding ECM and focal adhesion proteins, integrins are also signaling centers, transducing biochemical signals. Cell adhesion, or integrin clustering with antibodies or beads coated with ECM, rapidly activates Src-family tyrosine kinases (SFKs) and focal adhesion kinase (FAK), leading to tyrosine phosphorylation of several integrin-associated proteins and activating the GTPase Rac1 (*Burridge and Chrzanowska-Wodnicka, 1996*; *Parsons et al., 2010*). These signaling events are clearly important for regulating cell motility, cell cycle and cell survival, but their roles in focal adhesion dynamics remain unclear (*Burridge and Chrzanowska-Wodnicka, 1996*; *Mitra and Schlaepfer, 2006*).

One of the main substrates for integrin-stimulated tyrosine phosphorylation is an adaptor protein named Cas (p130Cas or BCAR1) (*Chodniewicz and Klemke, 2004*; *Janoštiak et al., 2014b*; *Mitra and Schlaepfer, 2006*). At the molecular level, Cas contains an N-terminal SH3 domain, a four-helix bundle, and a C-terminal FAT domain, separated by unstructured regions and an SFK SH3/SH2-binding site. Cas localizes to focal adhesions through its SH3 and FAT domains (*Donato et al., 2010*; *Nakamoto et al., 1997*), which bind vinculin, FAK, and paxillin in vitro (*Janoštiak et al., 2014a*; *Polte and Hanks, 1995*; *Zhang et al., 2017*). Cas and SFKs mutually activate each other, with Cas binding to and activating SFKs and SFKs phosphorylating Cas at up to 15 repeated YxxP motifs in the 'substrate domain' (SD) between the SH3 domain and four-helix bundle (*Chodniewicz and Klemke, 2004*; *Pellicena and Miller, 2001*). The Cas SD is also phosphorylated rapidly during cell adhesion (*Miyamoto et al., 1995*; *Petch et al., 1995*; *Vuori and Ruoslahti, 1995*). The trigger for Cas phosphorylation is unclear: integrin clustering or conformation changes or protein binding to the Cas SD may be involved (*Arias-Salgado et al., 2003*; *Hotta et al., 2014*; *Sawada et al., 2006*). After phosphorylation, the pYxxP motifs can bind specific SH2-domain proteins including the paralogs Crk and CrkL. Crk/CrkL in turn can bind to and stimulate various proteins, including the Rac1 guanine nucleotide exchange factor (GEF) DOCK180 (*Chodniewicz and Klemke, 2004*; *Gotoh et al., 1995*; *Hasegawa et al., 1996*; *Klemke et al., 1998*; *Knudsen et al., 1994*; *Tanaka et al., 1994*). DOCK180 can then activate Rac1. Rac1 promotes actin polymerization and lamellipodial protrusion through the WAVE/Arp2/3 complex, and induces focal complex formation through unknown mechanisms (*Nobes and Hall, 1995*; *Stradal et al., 2004*; *Zaidel-Bar et al., 2003*). Cas activity in focal adhesions is limited by the ubiquitin-proteasome system, which targets phosphorylated Cas for ubiquitination and degradation (*Steenkiste et al., 2021*; *Teckchandani and Cooper, 2016*; *Teckchandani et al., 2014*).

The role of integrin-activated tyrosine phosphorylation in focal adhesion dynamics is unclear. Early studies of Cas knockout mouse embryo fibroblasts (MEFs) revealed defects in cell attachment and the actin cytoskeleton, suggesting that Cas may regulate adhesion assembly (*Honda et al., 1999*; *Honda et al., 1998*). In addition, Cas regulates spreading and migration of Caco-2 epithelial cells (*Sanders and Basson, 2005*). However, other studies, using mutant MEFs lacking Cas, SFKs, FAK, or paxillin, revealed no change in adhesion assembly but major inhibition of adhesion disassembly (*Bockholt and Burridge, 1995*; *Ilić et al., 1995*, *Webb et al., 2004*). To revisit the role of Cas phosphorylation, we have studied focal adhesion assembly in spreading and migrating epithelial cells. Unexpectedly, we found that phosphorylated Cas co-clusters with inactive integrins nearly a minute before integrin activation and recruitment of core focal adhesion proteins. Cas is required for vinculin recruitment but vinculin is not required for Cas clusters to form. A positive feedback loop between SFKs, Cas, Crk, Rac1, and reactive oxygen species (ROS) promotes the growth of the early Cas–integrin clusters and subsequent integrin activation and focal adhesion assembly, opposed by negative feedback from the ubiquitin–proteasome system. The results suggest a key role for SFK–Cas–Crk–Rac1 signaling in early stages of focal adhesion formation.

# Results

## Cas clusters are precursors for vinculin clusters

Cas has the potential to serve as a signaling hub that may be critical for focal adhesion assembly. However, we are only aware of one study where Cas recruitment kinetics were measured relative to other focal adhesion proteins. The results showed that Cas and paxillin are recruited simultaneously during adhesion assembly in migrating fibroblasts (*Donato et al., 2010*). Another study using endothelial cells showed that tyrosine phosphorylation precedes paxillin recruitment (*Zaidel-Bar et al., 2003*). These two studies suggest that tyrosine phosphorylation may start before Cas is recruited. However, the kinetics of tyrosine phosphorylation, Cas recruitment, and adhesion assembly may vary according to cell type, integrin, or ECM. Therefore, we evaluated the kinetics of Cas recruitment and tyrosine phosphorylation relative to focal adhesion assembly, making use of the immortalized, normal, mammary epithelial line MCF10A (*Debnath and Brugge, 2005*). Since Cas over-expression can stimulate cell migration (*Klemke et al., 1998*; *Yano et al., 2000*), we tagged Cas by editing the *Cas* gene, inserting an artificial exon encoding mScarlet (mSc) and a linker sequence into the first intron and selecting a polyclonal population of mScarlet fluorescent cells (*Figure 1A*, *Figure 1—figure supplement 1*). Western blotting revealed similar levels and phosphorylation of Cas and Cas$^{mSc}$ proteins, indicating that most cells are heterozygous (*Figure 1B*). This intron tagging approach avoided the need for single-cell cloning that can select for variants (see Methods). To monitor focal adhesions, the Cas$^{mSc}$ cells were transduced to express near-endogenous levels of YFP-tagged vinculin (VCL). Imaging using total internal reflection (TIRF) microscopy revealed that Cas$^{mSc}$ and YFP-VCL substantially co-localized, as expected (*Figure 1C*). Staining with Cas and vinculin antibodies revealed that tagging Cas and vinculin did not alter focal adhesion number or structure (*Figure 1—figure supplement 2A, B*).

To compare Cas$^{mSc}$ and YFP-VCL dynamics, we performed dual-channel time-lapse TIRF imaging as cells attached and spread on collagen I (COLI). Cas$^{mSc}$ formed clusters at the first points of cell–substrate contact and moved outwards with the spreading edge (*Figure 1D*, *Video 1*). Vinculin joined these clusters later, and remained after Cas departed (note transition from magenta to white to green in kymographs). Similar patterns of Cas and vinculin clustering were observed under spontaneous lamellipodial protrusions generated by fully spread cells (*Figure 1E*, *Video 2*). These results together suggest that Cas clusters are precursors of vinculin clusters during spreading and migration.

To quantify the dynamics of cluster formation and avoid possible selection bias, we developed a computational pipeline to delineate regions of interest (ROIs) in which Cas, vinculin, or both Cas and vinculin intensities exceeded thresholds that were automatically set for each frame (see Methods). ROIs were tracked over time and quantified if they exceeded 20 pixels (0.5 μm$^2$) in area and persisted for three or more frames ($\geq$40 s) (*Figure 1F*, *Video 3*). These thresholds exclude the smallest, shortest-lived nascent adhesions but include larger focal complexes (*Kanchanawong and Calderwood, 2022*). The mean intensity of each channel in each ROI was then quantified over the duration of the recording and the intensities smoothed and normalized to a range of 0–1 (*Figure 1—figure supplement 3A*). For each ROI, we defined $\Delta t_{1/2}$ (VCL-Cas) as the time interval between Cas and vinculin reaching half-maximal intensity (*Figure 1—figure supplement 3B*). This metric showed wide variability in time interval across different ROIs in an individual cell. However, non-parametric analysis showed that the median $\Delta t_{1/2}$ (VCL-Cas) was significantly greater than zero (i.e., Cas clustering preceded vinculin clustering) (median $\Delta t_{1/2}$ 54.8 s, 95% CI 23–105 s, for 90 ROIs in the cell shown) (*Figure 1—figure supplement 3C*). Averaging the median time delay across multiple spreading cells in several experiments yielded a mean $\Delta t_{1/2}$ (VCL-Cas) 43.8 ± 3 s (mean and standard error of the mean [SEM], *n* = 19 cells) (*Figure 1G*). Similar results were obtained quantifying Cas and vinculin intensities under lamellipodia generated by migrating cells (*Figure 1—figure supplement 3D*). Averaging the median time delay across multiple migrating cells in several experiments yielded a mean $\Delta t_{1/2}$ (VCL-Cas) 68.5 ± 7 s (mean and SEM, *n* = 13 cells) (*Figure 1H*). A similar experiment using Cas$^{mSc}$ YFP-VCL HeLa cells also showed vinculin clustering after Cas (median $\Delta t_{1/2}$ (VCL-Cas) 48.1 ± 9 s, mean and SEM, *n* = 13 cells) (*Figure 1—figure supplement 3E, F*).

Overall, these results show a strong tendency for Cas to form clusters at the edge of spreading or migrating epithelial cells, 45–60 s before vinculin recruitment. For comparison, the time interval between arrival of talin, vinculin, and paxillin in nascent adhesions of CHO-K1 cells migrating on fibronectin (FN) is ~14 or ~2 s for non-maturing or maturing adhesions, respectively (*Han et al., 2021*).

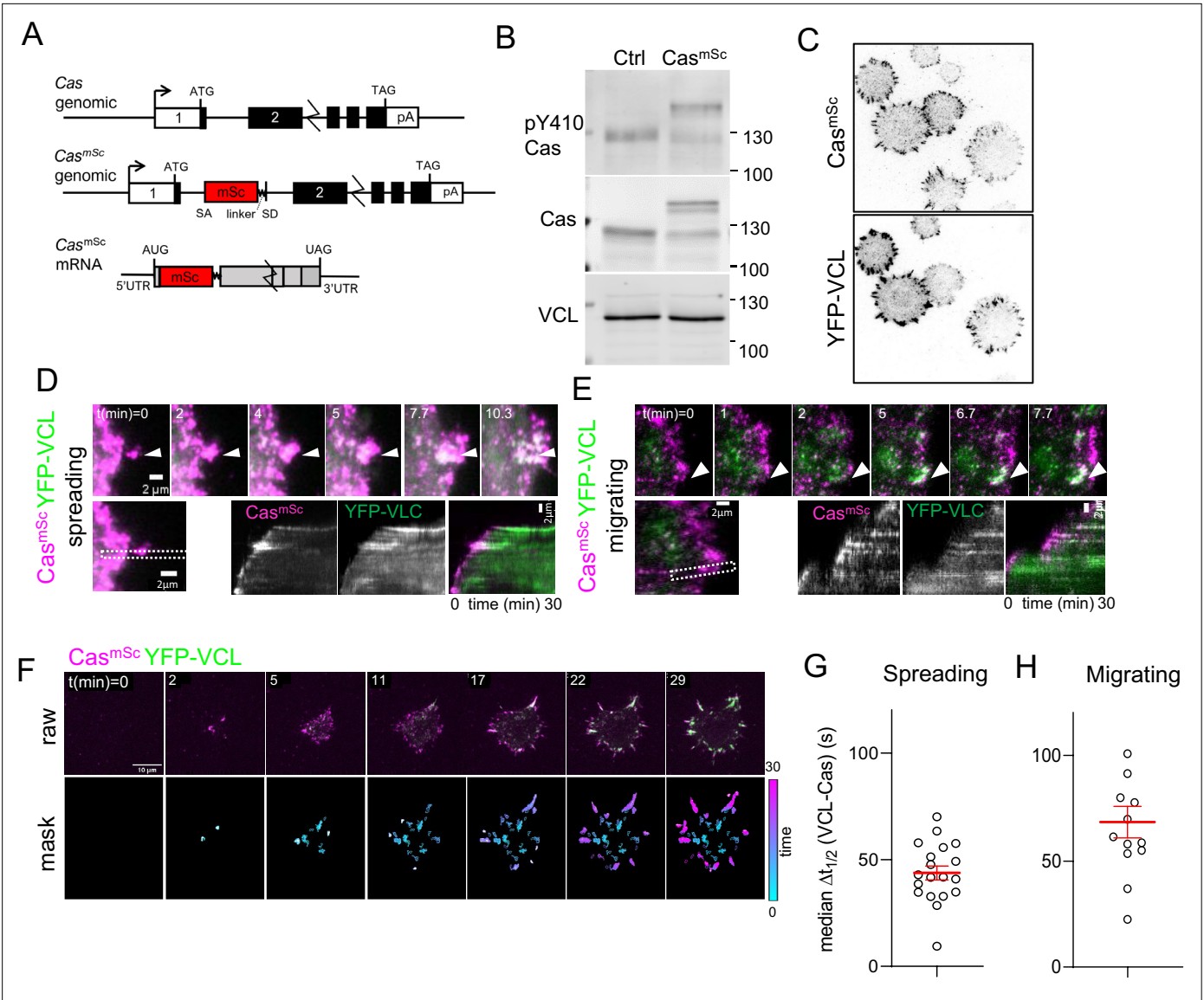

**Figure 1.** Cas clusters before vinculin during focal adhesion assembly. (**A**) *Cas* wildtype and *Cas^mSc* genomic organization and *Cas^mSc* mRNA structure. An artificial exon encoding mScarlet (mSc) and a 8-residue linker were inserted in intron 1. (**B**) Representative immunoblot showing the pY410Cas, total Cas and vinculin (VCL) in control (Ctrl) and Cas^mSc MCF10A cells. (**C**) Cas^mSc co-localization with YFP-VCL. Cas^mSc MCF10A cells expressing YFP-VCL were plated on COLI for 30 min and visualized by total internal reflection (TIRF) microscopy. (**D–H**) Cas^mSc clusters form before vinculin clusters. TIRF microscopy of Cas^mSc YFP-VCL cells. Individual time frames and kymographs from (**D**) spreading or (**E**) migrating cells. Arrowheads indicate a Cas^mSc clusters (magenta) that are later joined by YFP-VCL (green). (**F**) Pipeline for tracking regions of interest (ROIs). Upper panels: raw images. Lower panels: masks showing tracked ROIs, color coded by time of onset. (**G**) Median $\Delta t_{1/2}$ (VCL-Cas) of multiple ROIs from $n = 19$ spreading cells. Error bars show mean (43.8 s) and standard error of the mean (SEM) (3 s). (**H**) Median $\Delta t_{1/2}$ (VCL-Cas) of multiple ROIs from $n = 13$ spreading cells. Error bars show mean (68.5 s) and SEM (7 s).

The online version of this article includes the following figure supplement(s) for figure 1:

**Figure supplement 1.** Vector design and validation of Cas^mSc tagging.

**Figure supplement 2.** Tagging Cas and vinculin does not alter focal adhesion size or number.

**Figure supplement 3.** Pipeline for tracking Cas and vinculin cluster assembly.

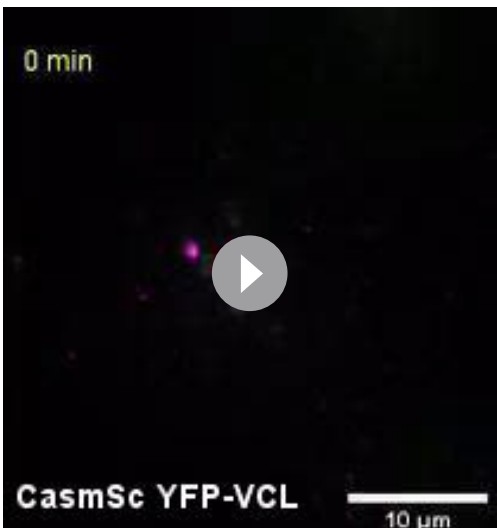

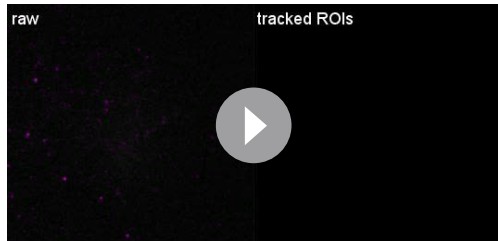

**Video 3.** Quantification of Cas and vinculin cluster dynamics. MCF10A CasmSc YFP-VCL MCF10A cell spreading on collagen. (Left) Raw data. (Right) Regions of interest defined as regions of 20 pixels (0.5 µm²) or greater in which either or both channel intensities exceed threshold in three consecutive frames. Regions are color coded according to the first frame in which the region is first detected, from cyan to magenta. 20 s time intervals.
https://elifesciences.org/articles/90234/figures#video3

The replacement of Cas clusters by vinculin clusters suggests that Cas may spatially coordinate vinculin clustering and adhesion assembly in both spreading and migrating MCF10A and HeLa cells.

**Video 1.** Cas (magenta) and vinculin (green) dynamics during attachment and spreading of a Cas^mSc YFP-VCL MCF10A cell on collagen. 15 s time intervals.
https://elifesciences.org/articles/90234/figures#video1

## Cas clusters are precursors of integrin clusters

To determine when integrins cluster relative to Cas, we transduced MCF10A Cas^mSc cells with a lentiviral vector encoding β1Ecto-pH, a recombinant integrin β1 with a pH-sensitive pHluorin tag inserted in the extracellular domain (*Huet-Calderwood et al., 2017*). This integrin is cell-surface expressed, localizes to adhesions, exhibits normal integrin activation, and restores adhesion in integrin β1 knockout MEFs (*Huet-Calderwood et al., 2017*). Live dual-color TIRF imaging of β1Ecto-pH in Cas^mSc MCF10A cells revealed that β1Ecto-pH localized to Cas clusters, but, like YFP-VCL, β1Ecto-pH kinetics were significantly delayed relative to Cas, with median $\Delta t_{1/2}$ (β1-Cas) 57.7 ± 6.7 s (mean and SEM, *n* = 19 cells) (*Figure 2A–C*, *Video 4* left).

As an independent approach to measure integrin clustering without ectopic expression, we tagged *ITGB1* by inserting an artificial exon into the *ITGB1* gene in HeLa Cas^mSc cells, adding an optimized linker (*Parsons et al., 2008*) and GFP tag to the C-terminus (*Figure 2D*, *Figure 2—figure supplement 1*). Western blotting showed a fusion protein of the expected mobility (*Figure 2D*). Live imaging Cas^mSc ITGB1^GFP cells revealed integrin clustering after Cas with median $\Delta t_{1/2}$ (ITGB1-Cas) 61.1 ± 5.4 s (mean and SEM, *n* = 15 cells) (*Figure 2F–H*, *Video 4* right). Thus, integrin β1 clusters about a minute after Cas in MCF10A and HeLa cells, at approximately the same time as vinculin.

## Initial Cas clusters contain Crk

Tyrosine phosphorylation is the earliest event during nascent adhesion formation in migrating fibroblasts (*Zaidel-Bar et al., 2003*). To determine when Cas is phosphorylated, we analyzed the recruitment of Crk, which binds to phosphorylated but not non-phosphorylated Cas and

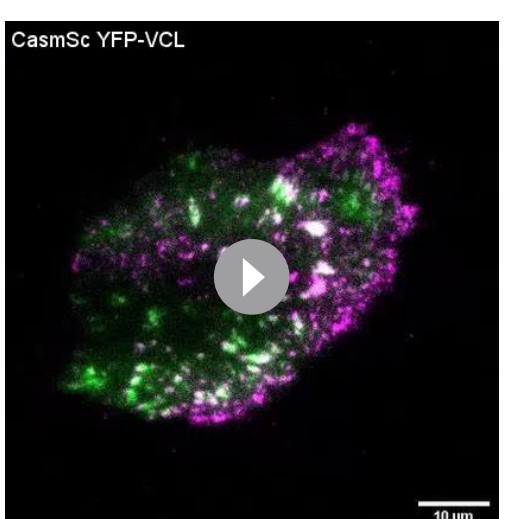

**Video 2.** Cas (magenta) and vinculin (green) dynamics during lamellipodia extension by a migrating Cas^mSc YFP-VCL MCF10A cell. 15 s time intervals.
https://elifesciences.org/articles/90234/figures#video2

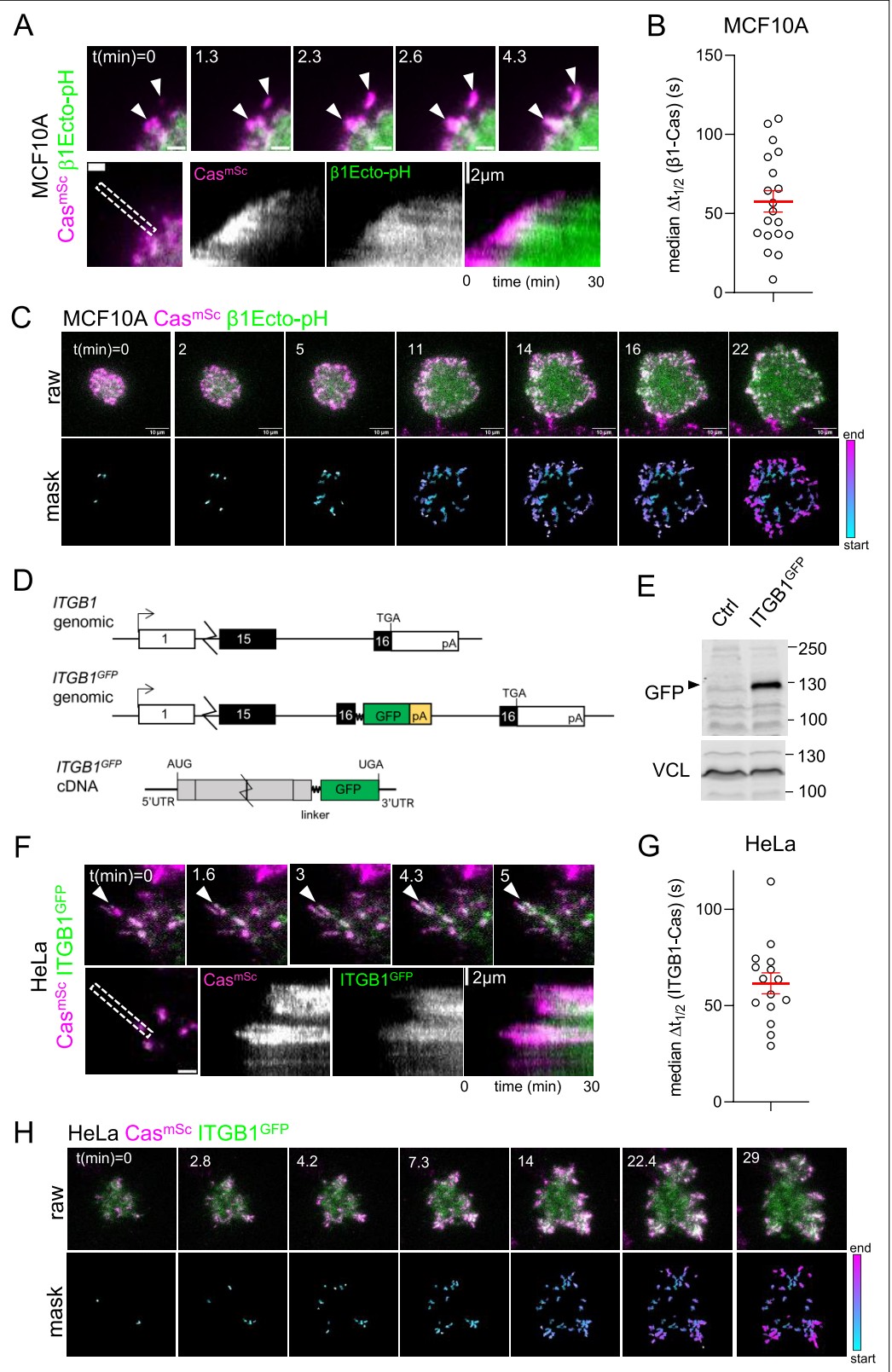

**Figure 2.** Cas clusters before integrin β1 during focal adhesion assembly. (**A**) Cas^mSc MCF10A cells were transduced to express β1Ecto-pH, plated on COLI and imaged 30 min by total internal reflection (TIRF) microscopy. Upper panels: individual time frames. Arrowheads indicate a Cas^mSc cluster (magenta) that is later joined by β1Ecto-pH (green). Lower panels: kymographs. (**B**) Median Δt_{1/2} (β1-Cas) of multiple regions of interest (ROIs) from 19

*Figure 2 continued on next page*

*Figure 2 continued*

spreading Cas[mSc] β1Ecto-pH MCF10A cells. Error bars show mean (57.7 s) and standard error of the mean (SEM) (6.7 s). (**C**) Upper panels: raw images. Lower panels: masks showing tracked ROIs, color coded by time of onset. (**D**) *ITGB1* wildtype and *ITGB1*[GFP] genomic organization and *ITGB1*[GFP] mRNA structure. An artificial exon encoding the ITGB1 C-terminus, linker, GFP, and polyA signal was inserted in intron 15. (**E**) Immunoblot showing expression of ITGB1[GFP] protein in Cas[mSc] ITGB1[GFP] HeLa cells. (**F**) Cas[mSc] clusters form before ITGB1[GFP] clusters. TIRF microscopy of Cas[mSc] ITGB1[GFP] HeLa cells. Upper panels: individual time frames. Arrowheads indicate a Cas[mSc] cluster (magenta) that is later joined by ITGB1[GFP] (green). Lower panels: kymographs. (**G**) Median $\Delta t_{1/2}$ (ITGB1-Cas) of multiple ROIs from 15 spreading Cas[mSc] ITGB1[GFP] HeLa cells. Error bars show mean (61.06 s) and standard error of the mean (SEM) (5.4 s). (**H**) Upper panels: raw images. Lower panels: masks showing tracked ROIs, color coded by time of onset.

The online version of this article includes the following figure supplement(s) for figure 2:

**Figure supplement 1.** Vector design and validation of ITGB1[GFP] tagging.

---

initiates downstream signaling (*Chodniewicz and Klemke, 2004*). We edited the *CRK* gene in Cas[mSc] HeLa cells, adding a linker and mGreenLantern (mGL) to the C terminus (*Figure 3A*, *Figure 3—figure supplement 1*). Western blotting showed approximately equal expression of Crk[mGL] and Crk, suggesting most cells are heterozygous (*Figure 3B*). Imaging Cas[mSc] Crk[mGL] cells during attachment and spreading revealed rapid recruitment of Crk[mGL] on Cas[mSc] clusters (*Video 5* left, *Figure 3C–E*). The median time delay, $\Delta t_{1/2}$ (Crk-Cas), was 5.7 ± 2.5 s (mean and SEM, *n* = 21 cells) (*Figure 3D*), much shorter than the time for vinculin or integrin recruitment. Crk recruitment was significantly delayed by treatment with eCF506, which inhibits all SFKs but not other kinases (*Fraser et al., 2016*), indicating that Crk recruitment requires phosphorylation (*Figure 3E*, *Video 5* right). Together these results suggest Cas is activated immediately as it first clusters, when integrin and vinculin density is still low.

## Spatial distribution of adhesion proteins in Cas–vinculin clusters

Time-lapse imaging of spreading Cas[mSc] YFP-VCL cells showed that Cas[mSc] is continuously added to the outer edge of Cas–vinculin clusters while YFP-VCL is added later, farther from the edge (*Figure 4A*). We confirmed this distribution by quantifying mScarlet and YFP intensity profiles along the axis of multiple, 4-µm-long Cas–vinculin clusters in several cells (*Figure 4B*). Cas[mSc] peaked ~0.75 µm and YFP-VCL peaked ~1.5 µm from the edge, reflecting their temporal order of recruitment. Similar profiles were obtained for endogenous Cas and vinculin when parental MCF10A cells were fixed and immunostained after 30 min of spreading (*Figure 4—figure supplement 1*). Thus, we reasoned that we could estimate the temporal order of arrival and departure of other adhesion proteins from their spatial distribution within Cas–vinculin clusters. To this end, spreading cells were fixed, immunostained with various combinations of antibodies, and imaged. Normalized intensity profiles were plotted and aligned using endogenous Cas or Cas[mSc] as a fiducial marker. Results are presented as heat maps in *Figure 4C* and sample images in *Figure 4—figure supplements 2 and 3*.

The intensity profiles show several important features. First, Cas phosphorylation, detected with pY410 Cas antibody, is maximal at the head of the cluster, ~0.5 µm from the cell edge (*Figure 4C*). We were unable to detect Crk with available antibodies but the Crk paralog CrkL peaked at the head of the cluster, consistent with high levels of phospho-Cas and rapid recruitment of Crk during adhesion assembly (*Figure 3C*). The head of the cluster also contained the Cas SH3-binding protein FAK, phosphorylated at Y861, an SFK phosphorylation site (*Eliceiri et al., 2002*). This population of FAK is unlikely to be active since it has low levels of autophosphorylation at Y397, a site required for kinase activity (*Le Coq et al., 2022*). This suggests that the head of the cluster is the peak of SFK activity.

Next, the peak of vinculin at ~1.5 µm also contains highest levels of mechanosensing and structural components of focal adhesions, including kindlin2, talin1, paxillin, kinase-active

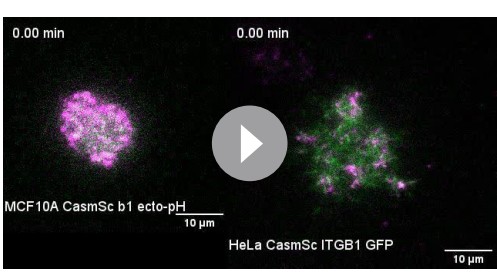

**Video 4.** Integrin clustering. Cas (magenta) and integrin (green) dynamics during attachment and spreading. Left: Cas[mSc]β1Ecto-pH MCF10A cell (20 s time intervals). Right: Cas[mSc] ITGB1[GFP] HeLa cell (15 s time intervals).

https://elifesciences.org/articles/90234/figures#video4

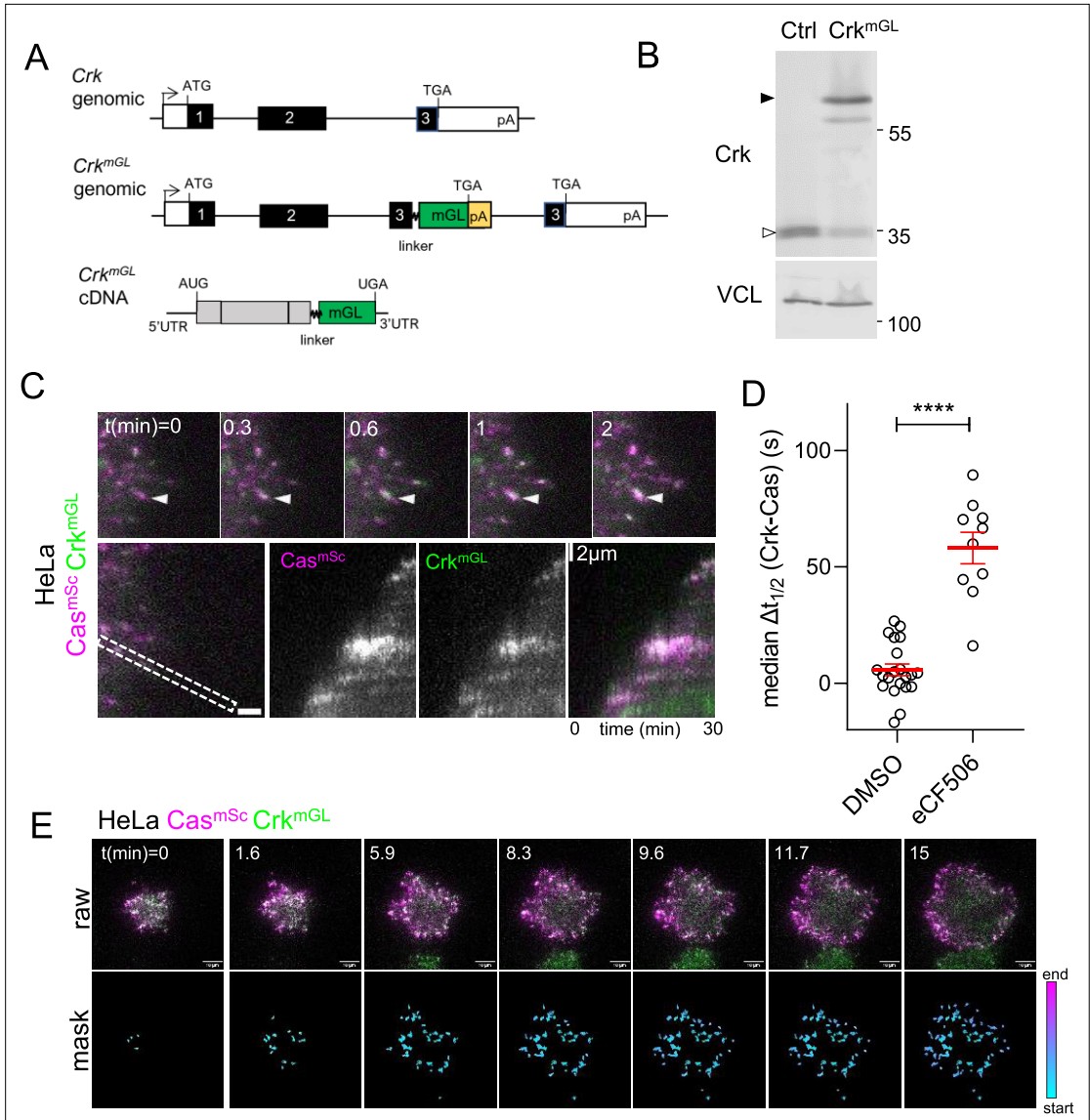

**Figure 3.** Cas and Crk cluster together. (**A**) *Crk*WT and *Crk*mGL genomic organization and *Crk*mGL mRNA structure. An artificial exon encoding the Crk C-terminus, linker, mGreenLantern (mGL), and polyA signal was inserted in intron 2. (**B**) Immunoblot showing expression of CrkmGL protein in CasmSc CrkmGL HeLa cells. (**C**) CrkmGL clusters form shortly after CasmSc clusters. Total internal reflection (TIRF) micrographs of CasmSc CrkmGL HeLa cells. Upper panels: individual time frames. Arrowheads indicate a CasmSc cluster (magenta) that is rapidly joined by CrkmGL (green). Lower panels: kymographs. (**D**) Median $\Delta t_{1/2}$ (Crk-Cas) of multiple regions of interest (ROIs) from 10 to 21 spreading CasmSc CrkmGL HeLa control and eCF506-treated cells. Error bars show mean and standard error of the mean (SEM). ***,p < 0.001 by Mann–Whitney *U*-test. (**E**) Upper panels: raw images. Lower panels: masks showing tracked ROIs, color coded by time of onset.

The online version of this article includes the following figure supplement(s) for figure 3:

**Figure supplement 1.** Vector design and validation of CrkmGL tagging.

pY397 FAK, and F-actin, but decreased amounts of pY410Cas and total Cas (*Figure 4C*). The presence of mechanosensing proteins suggests this part of the cluster is attached to the ECM and subject to mechanical force.

Finally, the distribution of integrin β1 varied according to the antibody used. Total integrin β1, detected with conformation-insensitive antibody AIIB2 (*Mould et al., 2016*), forms two peaks, one coincident with pY410Cas and pY861FAK and one aligning with the peak of vinculin. The first peak was also detected with mAb13, specific for inactive, bent-closed (BC) and extended-closed (EC) β1 integrin conformations (*Su et al., 2016*). The second peak was detected with 12G10, specific for the

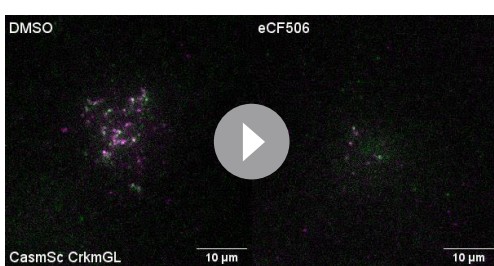

**Video 5.** Cas–Crk dynamics. Cas (magenta) and Crk (green) dynamics during attachment and spreading of Cas$^{mSc}$ Crk$^{mGL}$ HeLa cells in presence of dimethylsulfoxide (DMSO) (left) or SFK inhibitor eCF506 (right). 20 s time intervals.
https://elifesciences.org/articles/90234/figures#video5

active, extended-open (EO) conformation, and by 9EG7, which detects both EO and EC conformations (*Su et al., 2016*; *Figure 4C, D*). This suggests that integrin activation increases as kindlin, talin, and vinculin are recruited, consistent with the roles of kindlin/talin binding and mechanical force in integrin activation.

Overall, while differential antibody access may affect immunofluorescent staining profiles, the spatial patterns suggest a temporal sequence of events in which active phosphorylated Cas and Crk/CrkL initially cluster with inactive integrin β1 and SFK phosphorylated but kinase-inactive FAK. Since integrin β1 is in its BC conformation it is probably not attached to the ECM. Later, phosphorylated and total Cas levels decrease as integrin β1 recruits talin and kindlin, adopts the EO conformation, and able to bind ECM. Mechanical forces from the actin cytoskeleton then expose binding sites for vinculin and a mature focal adhesion is formed (model, *Figure 4E*).

## Phosphorylated Cas is enriched in the adhesome

Even though many imaging studies have detected Cas in focal adhesions, Cas is not routinely detected in the adhesome, as defined by proteomics or proximity biotinylation (*Kanchanawong and Calderwood, 2022*). However, low abundance proteins may be significantly enriched in the adhesome yet escape detection. We used Western blotting to estimate the proportion of Cas and selected other adhesion proteins in the adhesome relative to the non-adhesome (supernatant) fraction. Cells were seeded on polylysine or COLI for 60 min and incubated with protein–protein cross-linking reagents before lysis and separation of adhesome and supernatant fractions as described (*Humphries et al., 2009*; *Schiller et al., 2011*). We then reversed the cross-links and performed Western blotting on the adhesome and supernatant samples. We found that the adhesome contained ~20–27% of total cellular integrin β1, talin, vinculin, paxillin, and total and autophosphorylated (pY397) FAK, but only ~7% of a control protein, ERK (*Figure 4—figure supplement 4*). The adhesome fraction also contained ~25% of total cellular Cas, suggesting it is as enriched in adhesions as bona fide adhesome proteins. Moreover, the adhesome also contained ~73% of phospho-Cas, detected with pY410 or pY249 antibodies, and pY861 FAK. This suggests that pY410 Cas, pY249 Cas, and pY861 FAK are significantly enriched in the adhesome fraction relative to their non-phosphorylated (or in the case of FAK, pY397 phosphorylated) forms. This suggests that these phosphorylations occur locally within adhesions and are rapidly lost when Cas or FAK dissociate. In contrast, autophosphorylated pY397 FAK remains active in the cytosol.

## Cas, Crk, and SFKs regulate vinculin recruitment and focal adhesion assembly

The early arrival of Cas at sites of future focal adhesions suggests Cas may nucleate adhesion assembly. To test this possibility, we depleted Cas from Cas$^{mSc}$ cells using siRNA (*Figure 5—figure supplement 1A*). Cas-depleted cells were non-migratory and their adhesions were immobile (*Figure 5—figure supplement 1B*, *Video 6*). Initial cell attachment and spreading were strongly inhibited, with significant decreases in cell area. Cas depletion inhibited focal adhesion formation, and Cas and vinculin intensity in remaining adhesions were inhibited to a similar degree (*Figure 5A, B*). Immunostaining of other focal adhesion proteins including talin1, kindlin2, and FAK in the remaining adhesions was also reduced (*Figure 5—figure supplement 1C*). By way of comparison, we treated cells with vinculin siRNA. Vinculin depletion was inefficient (*Figure 5—figure supplement 1A*), but significantly reduced vinculin intensity in adhesions (*Figure 5A, B*). However, there was no reduction in cell spreading, the number of Cas clusters, or the intensity of Cas in vinculin-depleted cells. This suggests that Cas regulates vinculin recruitment but vinculin does not regulates Cas recruitment, consistent with their order of assembly.

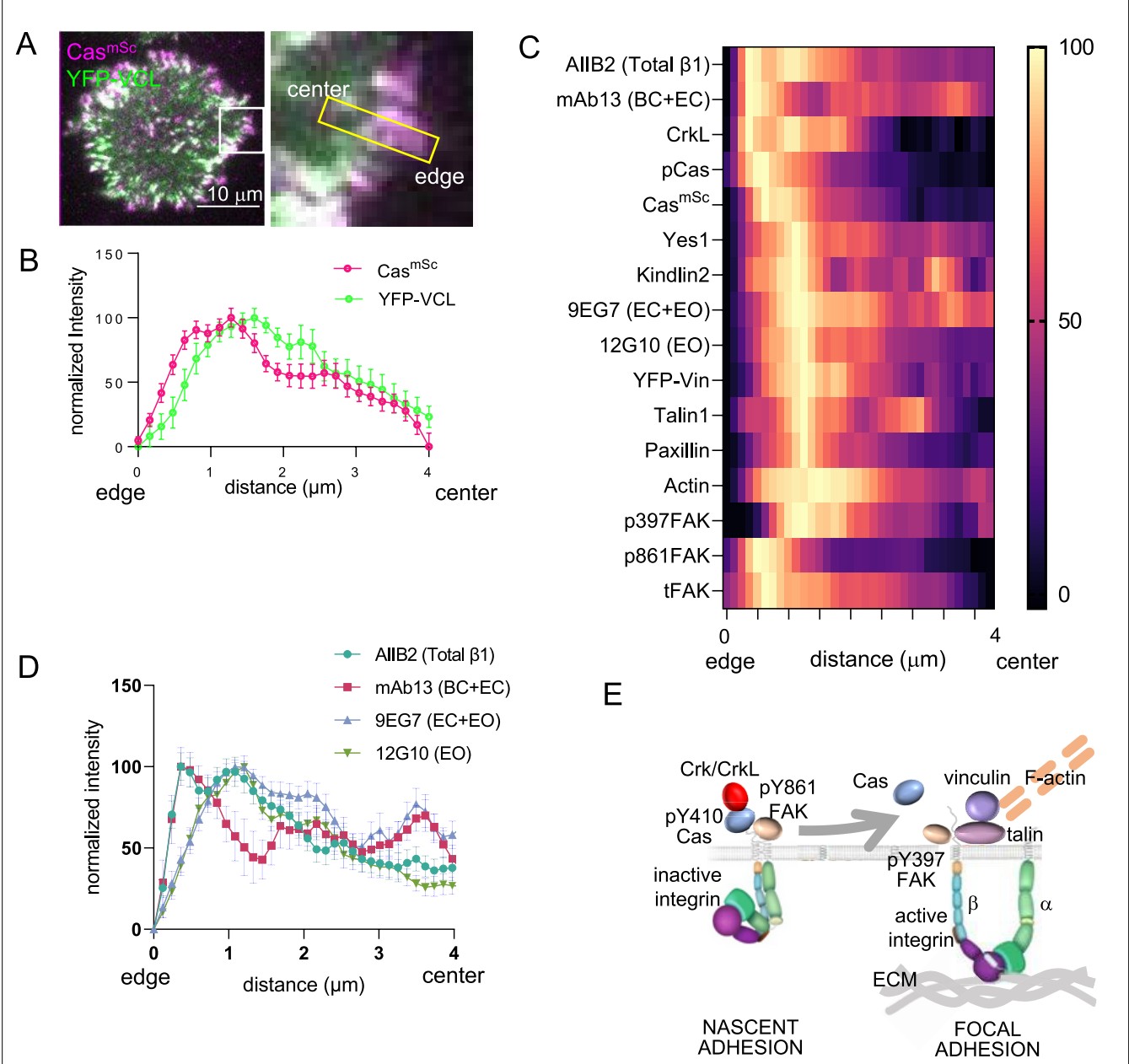

**Figure 4.** Spatial distribution of proteins in focal adhesions. (**A**) Representative image of spreading Cas^mSc YFP-VCL MCF10A cells illustrating quantification approach. Inset shows 4 × 0.8 µm region of interest (ROI) used to quantify intensity against distance from cell edge. (**B**) Normalized intensity profiles of Cas^mSc and YFP-VCL across ≥20 ROIs from several cells. Error bars indicate mean and standard error of the mean (SEM). (**C**) Heat map of normalized intensity profiles for various antigens in Cas^mSc cell adhesions stained with indicated antibodies. (**D**) Normalized intensity profiles using conformation-sensitive integrin β1 antibodies; AIIB2, total integrin β1; mAb13, bent closed (BC) and extended closed (EC) conformations; 9EG7, EC and extended open (EO) conformations; 12G10, EO conformation. (**E**) Model showing inferred progression from nascent adhesions or focal complexes containing high levels of pY410Cas, inactive integrin and pY861FAK, to focal adhesions containing low levels of Cas and high levels of active integrin, vinculin, pY397FAK, F-actin, talin, paxillin, and kindlin.

The online version of this article includes the following figure supplement(s) for figure 4:

**Figure supplement 1.** Spatial distribution of endogenous untagged Cas and vinculin in focal adhesions of parental MCF10A cells.

**Figure supplement 2.** Representative images quantified for *Figure 4*.

**Figure supplement 3.** Representative images quantified for *Figure 4*.

**Figure supplement 4.** Enrichment of phosphorylated Cas in the adhesome.

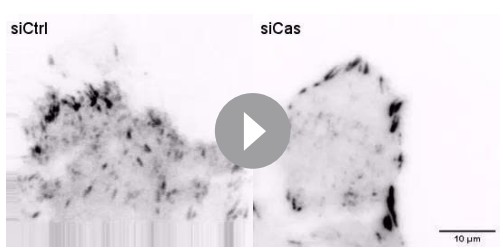

**Video 6.** Vinculin dynamics in control (left) and Cas-depleted (right) Cas^mSc^ YFP-VCL cells, 24 hr after plating on collagen. 20 s time intervals.
https://elifesciences.org/articles/90234/figures#video6

To control for possible non-specific effects of Cas siRNA, we transduced MCF10A cells to express tagged wildtype or mutant mouse Cas, and knocked down endogenous Cas with human Cas-specific siRNA. As shown in *Figure 5—figure supplement 2*, wildtype mouse Cas (mCasWT) rescued cell spreading and adhesion formation and was recruited approximately a minute before vinculin. However, mutant mouse Cas (mCas15F), lacking the fifteen YxxP phosphorylation sites in the SD, did not rescue cell spreading or adhesion formation.

We investigated whether Crk/CrkL and SFKs are required for Cas-dependent focal adhesion assembly. We used siRNA to deplete Crk, CrkL, or both and measured the time lag between Cas and vinculin recruitment. Depleting Crk and CrkL together but not separately slowed vinculin recruitment significantly, suggesting functional overlap (*Figure 5C*, *Figure 5—figure supplement 3A–C*, *Video 7*). Inhibiting SFKs with the pan-SFK inhibitor eCF506 also significantly delayed vinculin recruitment (*Figure 5D*, *Video 7* bottom panels). To test whether a specific SFK is required, we knocked down each of the major SFKs – Src, Fyn, and Yes1 – and measured adhesion and spreading. Remarkably, cell spreading, adhesion number, vinculin intensity, and Cas phosphorylation were all inhibited by depletion of Yes1 but not Src or Fyn, while Cas cluster formation was normal (*Figure 5E*, *Figure 5—figure supplement 3D–F*). This implies that Yes1 may have a special role in phosphorylating Cas and recruiting Crk/CrkL to stimulate adhesion assembly in MCF10A cells spreading.

## FAK, kindlin, and talin are not required for Cas clustering

We found that Cas and FAK cluster together early during adhesion assembly (*Figure 4C*), raising the possibility that FAK recruits Cas, as reported in spreading fibroblasts (*Zhang et al., 2014*). Therefore, we tested whether FAK is required for Cas clustering by depleting FAK with siRNA. Depleting FAK had no effect on MCF10A cell spreading, the intensity of Cas or vinculin clusters, or the median time delay between Cas and vinculin recruitment (*Figure 5—figure supplement 4A–C*). Kindlin2 and talin1 directly bind integrin tails and play important roles in integrin adhesion in fibroblasts, with kindlin2 interacting with paxillin and talin1 recruiting vinculin and providing attachment for actin, enabling force generation, adhesion maturation and cell spreading (*Bachmann et al., 2023*). Kindlin2 or talin1 siRNA had no effect on Cas clustering, but depleting talin1 decreased cell spreading and the intensity of vinculin clusters, and depleting either kindlin2 or talin1 increased the time delay between Cas and vinculin recruitment (*Figure 5—figure supplement 4D–F*). This suggests a sequence of events where Cas regulates cell spreading and recruitment of talin1 and kindlin2, and talin1 and kindlin2 are required to recruit vinculin.

## Cas regulates 'outside-in' integrin activation

Integrins can be activated 'inside-out' by proteins binding to their cytoplasmic tails, or 'outside-in' by interactions with the ECM or with molecules or ions that stabilize the active open conformation (*Hynes, 2002*). We tested whether Cas is required for outside-in adhesion assembly by treating control or Cas-depleted cells with $Mn^{2+}$, which stabilizes the active integrin conformation (*Gailit and Ruoslahti, 1988*; *Lenter et al., 1993*). While $Mn^{2+}$ increased the spread area of control cells, it did not increase spreading of Cas siRNA-treated cells and did not rescue the intensity of vinculin clusters (*Figure 5F*). As a control for potential off-target effects of Cas siRNA we found that expression of mCasWT but not mCas15F rescued outside-in signaling by $Mn^{2+}$ (*Figure 5—figure supplement 5A–C*). This suggests that phosphorylation of Cas is rate limiting for outside-in as well as inside-out integrin activation.

## Cas regulates focal adhesion assembly on different ECM, in different cell types, and through different integrins

Our finding that Cas and SFKs are required for adhesion assembly conflicts with previous results (*Bockholt and Burridge, 1995*; *Webb et al., 2004*). However, the previous studies used mutant

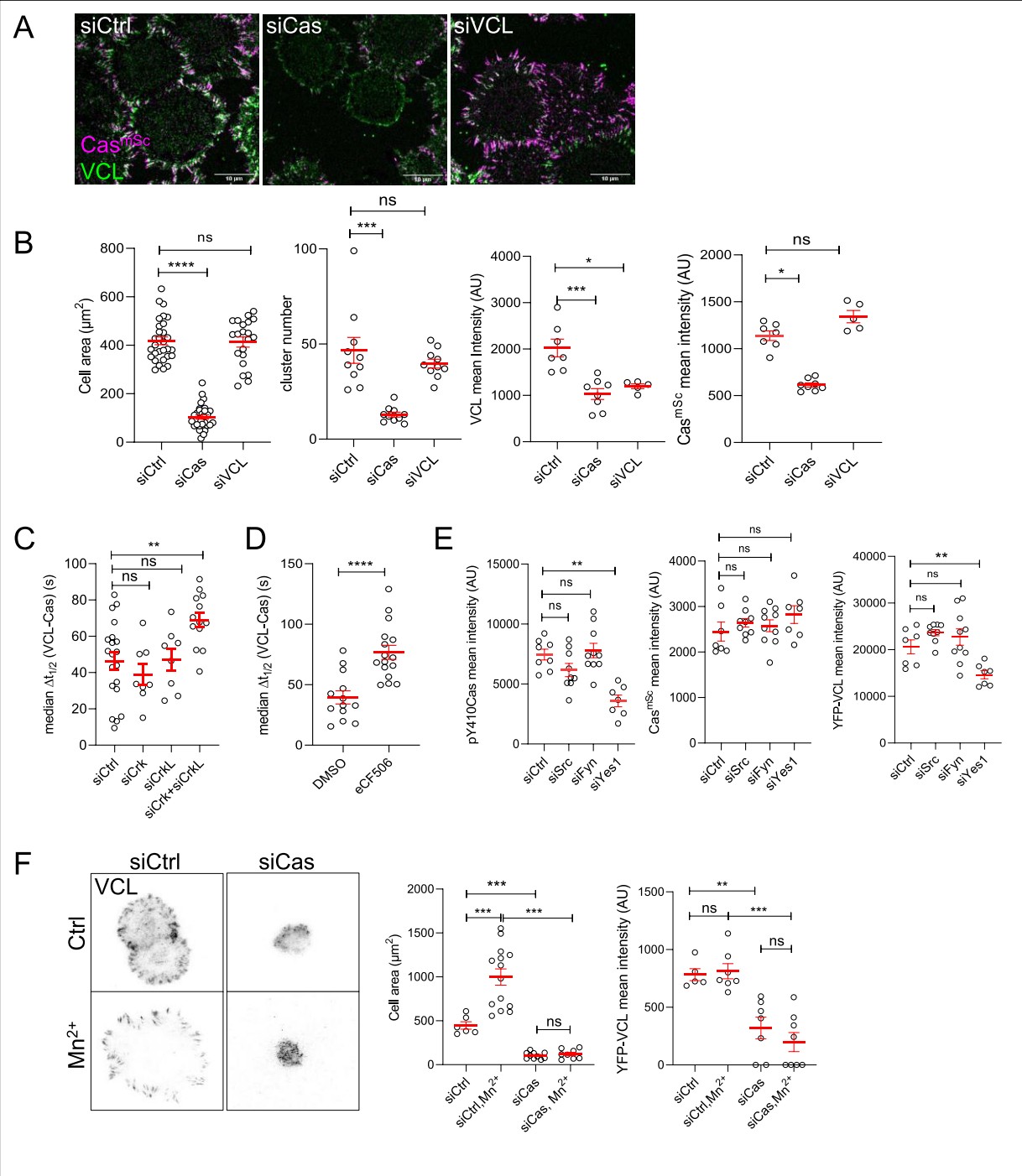

**Figure 5.** Cas is required for focal adhesion assembly. (**A**) Representative images (ventral section) of Cas^mSc MCF10A cells treated with control, Cas, or vinculin siRNA and fixed and stained with vinculin antibodies after 30 min of spreading on COLI. (**B**) Quantification of mean cell area and the number and mean intensities of Cas and/or vinculin clusters. Error bars show mean and standard error of the mean (SEM) for $n$ = 10–50 cells from three biological repeats. (**C**) Median $\Delta t_{1/2}$ (VCL-Cas) of multiple regions of interest (ROIs) from 8 to 20 spreading Cas^mSc YFP-VCL MCF10A cells treated with Ctrl, Crk, CrkL, and Crk + CrkL siRNA. Error bars show mean and SEM. (**D**) Median $\Delta t_{1/2}$ (VCL-Cas) from 13 to 16 time-lapse dual-color total internal reflection (TIRF) micrographs of spreading Cas^mSc YFP-VCL MCF10A cells treated with DMSO or SFK inhibitor eCF506. (**E**) Mean cluster intensity of pY410Cas, Cas^mSc and YFP-VCL in Cas^mSc YFP-VCL MCF10A cells treated with control, Src, Fyn, or Yes1 siRNA and fixed after 30 min of spreading. Error bars show mean and SEM from $n$ = 7–10 cells from three biological repeats. (**F**) Cas requirement for outside-in signaling. YFP-VCL MCF10A cells were treated with control or Cas siRNA and allowed to attach in the absence or presence of $Mn^{2+}$ for 30 min. Graphs show the mean cell spread area and mean intensity of YFP-VCL clusters. Error bars show mean and SEM for $n$ = 6–20 cells from two biological repeats. ns, not significant; *p < 0.05; **p < 0.01; ***p < 0.001; ****p < 0.0001 by Kruskal–Wallis followed by Dunn's multiple comparison test (**A–E**) or pairwise Mann–Whitney $U$-tests (**F**).

*Figure 5 continued on next page*

*Figure 5 continued*

The online version of this article includes the following figure supplement(s) for figure 5:

**Figure supplement 1.** Cas depletion inhibits cell migration, spreading, and formation of adhesions containing focal adhesion kinase (FAK), talin1, and kindlin2.

**Figure supplement 2.** Cell spreading and adhesion assembly require Cas phosphorylation sites.

**Figure supplement 3.** SFK–Cas–Crk–Rac1 signaling regulates adhesion assembly.

**Figure supplement 4.** Focal adhesion kinase (FAK), talin1, and kindlin2 do not regulate Cas clusters.

**Figure supplement 5.** Outside-in integrin activation requires Cas phosphorylation sites.

---

fibroblasts spreading on FN while we used epithelial cells on COLI. This raises the possibility that Cas may be dispensable for adhesion formation in certain cell types or ECM. To investigate further, we depleted Cas from MCF10A cells and plated them on FN or COLI. Cas depletion inhibited cell spreading and the number, area and vinculin intensity of adhesions on both substrates (*Figure 6A*). Moreover, time-lapse imaging showed that Cas preceded vinculin clustering by a similar time interval on FN as on COLI (*Figure 6B*, *Video 8*). Therefore, Cas plays a similar role in adhesion assembly when epithelial cells attach to either FN or COLI.

To test whether Cas also regulates adhesion assembly in fibroblasts, we depleted Cas from human foreskin fibroblasts (HFFs) and plated them on FN or COLI. As with epithelial cells, Cas depletion inhibited HFF cell spreading and adhesion formation on both ECMs (*Figure 6C*, *Figure 6—figure supplement 1A*).

Proteomic analysis of MCF10A cells reveals expression of a variety of α and β integrin chains (*Ly et al., 2018*). However, all α chains that are expressed can heterodimerize with β1, and all the β chains that are expressed can heterodimerize with αv (*Hynes, 2002*). We tested the roles of β1 and αv heterodimers in Cas-dependent adhesion of epithelial and fibroblast cells using siRNA. On COLI, β1 depletion strongly inhibited spreading and recruitment of vinculin to Cas clusters, while depleting αv had smaller effects. In contrast, on FN, both β1 and αv were required for spreading and vinculin recruitment (*Figure 6—figure supplement 1B, C*). Thus, Cas regulates β1-dependent adhesion assembly on COLI, and β1- or αv-dependent adhesion assembly FN. To further identify the specific integrins regulated by Cas, we noted that MCF10A cells express α chains α2, 3, 5, and v at 10-fold higher level than other α chains, while β3 is under-expressed relative to β1 and β6 (*Ly et al., 2018*). This makes α2β1 a strong candidate to bind COLI and α5β1 and αvβ6 strong candidates to bind FN. Indeed, integrin α2β1-blocking mAb P1E6 (*Carter et al., 1990*; *Tuckwell et al., 1995*) prevented COLI binding while integrin α5β1-blocking monoclonal antibody (mAb) P8D4 (*Alfandari et al., 2003*; *Davidson et al., 2002*) prevented FN binding (*Figure 6—figure supplement 2A*). The exact αv integrin regulated by Cas is unclear, but immunofluorescence of HFFs spreading on FN did not reveal any αv clusters that do not also contain β1 (*Figure 6—figure supplement 2B*). Early β1 clusters at the edge lack αv. Taken together, these results suggest that Cas nucleates α2β1 and α5β1 adhesions on COLI and FN, respectively, and is also required for subsequent recruitment of αv on FN.

## SFK–Cas–Crk–Rac1 signaling regulates adhesion assembly

SFK–Cas–Crk/CrkL signaling is mediated by Crk/CrkL effectors including DOCK180, a Rac1 GEF (*Chodniewicz and Klemke, 2004*). Rac1 is known to regulate lamellipodia protrusion by binding to the WAVE regulatory complex (WRC) and driving Arp2/3 complex-mediated actin polymerization (*Takenawa and Suetsugu, 2007*). Transient activation of Rac1 also induces formation of nascent adhesions at the cell edge, although the specific mechanism is unclear (*Nobes and Hall, 1995*; *Zaidel-Bar et al., 2003*). We used a Forster resonance energy transfer (FRET) biosensor (Rac1-2G) to measure Rac1 activity during MCF10A cell

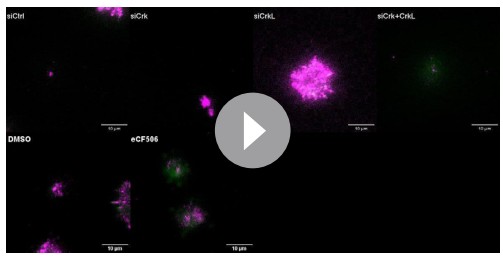

**Video 7.** Crk/CrkL and SFK requirement. Cas^mSc YFP-VCL MCF10A cells treated with siCtrl (upper left), siCrk, siCrkL, or siCrkL/CrkL (upper right) or with DMSO (bottom left) or eCF506 (bottom right). 20 s time intervals.
https://elifesciences.org/articles/90234/figures#video7

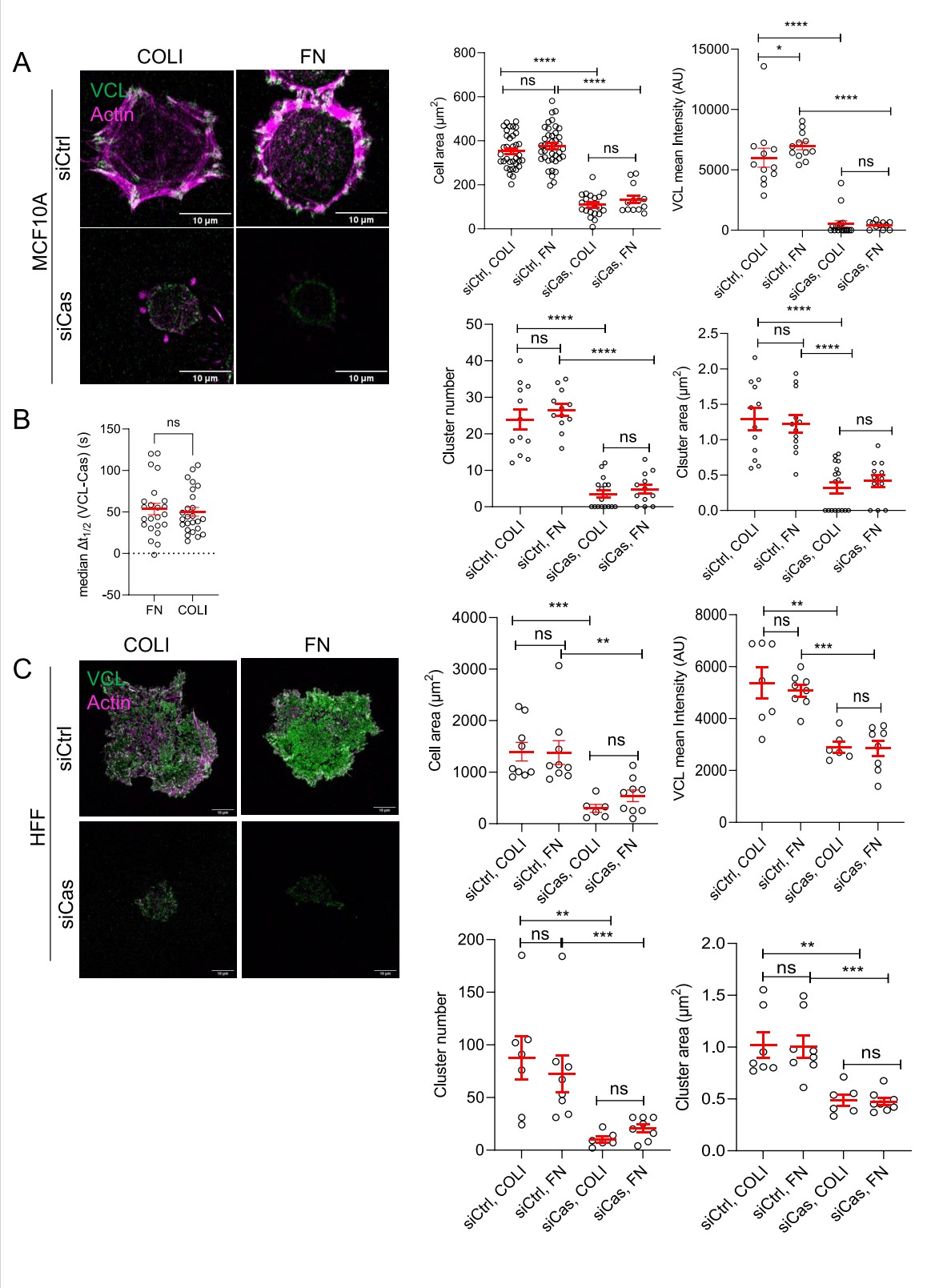

**Figure 6.** Cas is required for MCF10A and human foreskin fibroblast (HFF) cell spreading and adhesion assembly on different extracellular matrix (ECM). (**A**) MCF10A cells plated for 30 min on COLI- or fibronectin (FN)-coated surfaces after control or Cas siRNA treatment. Mean cell area, mean VCL intensity, mean cluster area and number of 7–41 cells in two biological repeats. (**B**) Median $\Delta t_{1/2}$ (VCL-Cas) of Cas$^{mSc}$ YFP-VCL MCF10A cells spreading on FN or COLI. Error bars indicate mean and standard error of the mean (SEM). $n$ = 23–25. ns, non-significant by Mann–Whitney test. (**C**) HFF cells plated

*Figure 6 continued on next page*

*Figure 6 continued*

for 30 min on COLI- or FN-coated surface after control or Cas siRNA treatment. Mean cell area, mean VCL intensity, mean cluster number, and area, of 6–9 cells in two biological repeats. ns, non-significant; *p < 0.05; **p < 0.01; ***p < 0.001; ****p < 0.0001 by Kruskal–Wallis followed by Dunn's multiple comparison test.

The online version of this article includes the following figure supplement(s) for figure 6:

**Figure supplement 1.** β1 integrin is required for adhesion on both COLI and fibronectin (FN), while αv is required for adhesion on FN.

**Figure supplement 2.** Requirements and organization of specific integrins.

adhesion (*Fritz et al., 2015*). Rac1-2G FRET activity was stimulated around the periphery of cells attaching to collagen and was inhibited by the Rac1-specific inhibitor EHT1864 (*Onesto et al., 2008*; *Figure 7A*). EHT1864 also inhibited cell spreading and reduced and delayed vinculin clustering but had no effect on Cas clustering (*Figure 7B*, *Video 9* top). Importantly, depleting Cas or Yes1 inhibited Rac1 (*Figure 7C, D*), consistent with Rac1 activation by the SFK–Cas–Crk/CrkL pathway (*Chodniewicz and Klemke, 2004*). Together, these results suggest that early clustering of phosphorylated Cas with Crk and inactive integrins activates Rac1 to trigger both cell spreading and assembly of vinculin-containing focal adhesions.

## Linkage between cell spreading and adhesion assembly through positive and negative feedback

If the sole function of Cas in focal adhesion assembly is to activate Rac1, then Rac1 activation may bypass the need for Cas. To test this possibility, we over-expressed either wildtype or constitutively active GFP-Rac1$^{Q61L}$ in MCF10A Cas$^{mSc}$ cells by transient transfection and examined cells during attachment to collagen. As expected, GFP-Rac1$^{Q61L}$ cells spread more than GFP-Rac1$^{WT}$ cells (*Figure 8A*). Surprisingly, Cas depletion prevented Rac1-induced spreading (*Figure 8A*). This suggests that, even though Cas requires Rac1 to support normal spreading and adhesion formation, active Rac1 requires Cas to induce spreading, raising the possibility of a positive feedback loop from Rac1 back to Cas.

We investigated which Rac1 effector may be involved in positive feedback on Cas. Rac1 stimulates localized production of ROS by the Nox1 NADPH-dependent oxidase by binding p47$^{phox}$ (*Ushio-Fukai, 2006*). ROS are short-range signaling molecules, rapidly reacting with and inhibiting protein-tyrosine phosphatases (PTPs) and increasing local tyrosine phosphorylation of various substrates including Cas, SFKs, and p190RhoGAP (*Garton et al., 1996*; *Giannoni et al., 2005*; *Nimnual et al., 2003*; *Tonks, 2005*). Thus, Rac1 potentially activates Cas through a Rac1–ROS–PTP–SFK–Cas pathway. Such a positive feedback loop, where the output signal is fed back as input, can amplify the signal, enhancing the outcome (*Brandman and Meyer, 2008*). To test whether ROS are involved in positive feedback and adhesion assembly in MCF10A cells, we inhibited Nox1 with diphenylamineiodonium (DPI) (*Reis et al., 2020*). DPI inhibited Rac1 (*Figure 8B*), inhibited cell spreading, and delayed vinculin recruitment to Cas clusters (*Figure 8C*, *Video 9*). This suggests that positive feedback through ROS amplifies and sustains SFK–Cas–Crk/CrkL–Rac1 activity that recruits vinculin.

Positive feedback loops require negative feedback to avoid runaway amplification (*Brandman and Meyer, 2008*). Phosphorylation-dependent signaling can be inhibited by PTPs or phosphorylation-dependent proteolysis. Cas signaling is inhibited by negative feedback through the CRL5$^{SOCS6}$ ubiquitin ligase complex, which targets activated Cas for proteasomal degradation (*Teckchandani and Cooper, 2016*; *Teckchandani et al., 2014*). We tested whether CRL5$^{SOCS6}$ regulates focal adhesion assembly during MCF10A cell attachment. Either SOCS6 depletion or a Cullin inhibitor, MLN4924, shortened $\Delta t_{1/2}$ (VCL-Cas) during cluster assembly, consistent with CRL5$^{SOCS6}$ interfering with focal adhesion formation by negative feedback on phospho-Cas signaling (*Figure 8D*, *Video 10*).

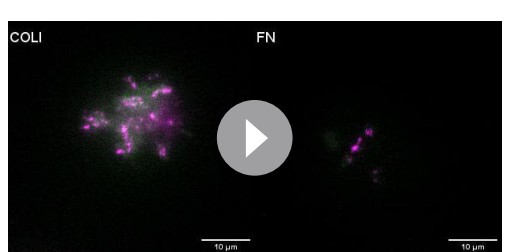

**Video 8.** Effect of extracellular matrix (ECM). Cas (magenta) and vinculin (green) dynamics during attachment and spreading on COLI (left) and fibronectin (FN; right). 20 s time intervals.

https://elifesciences.org/articles/90234/figures#video8

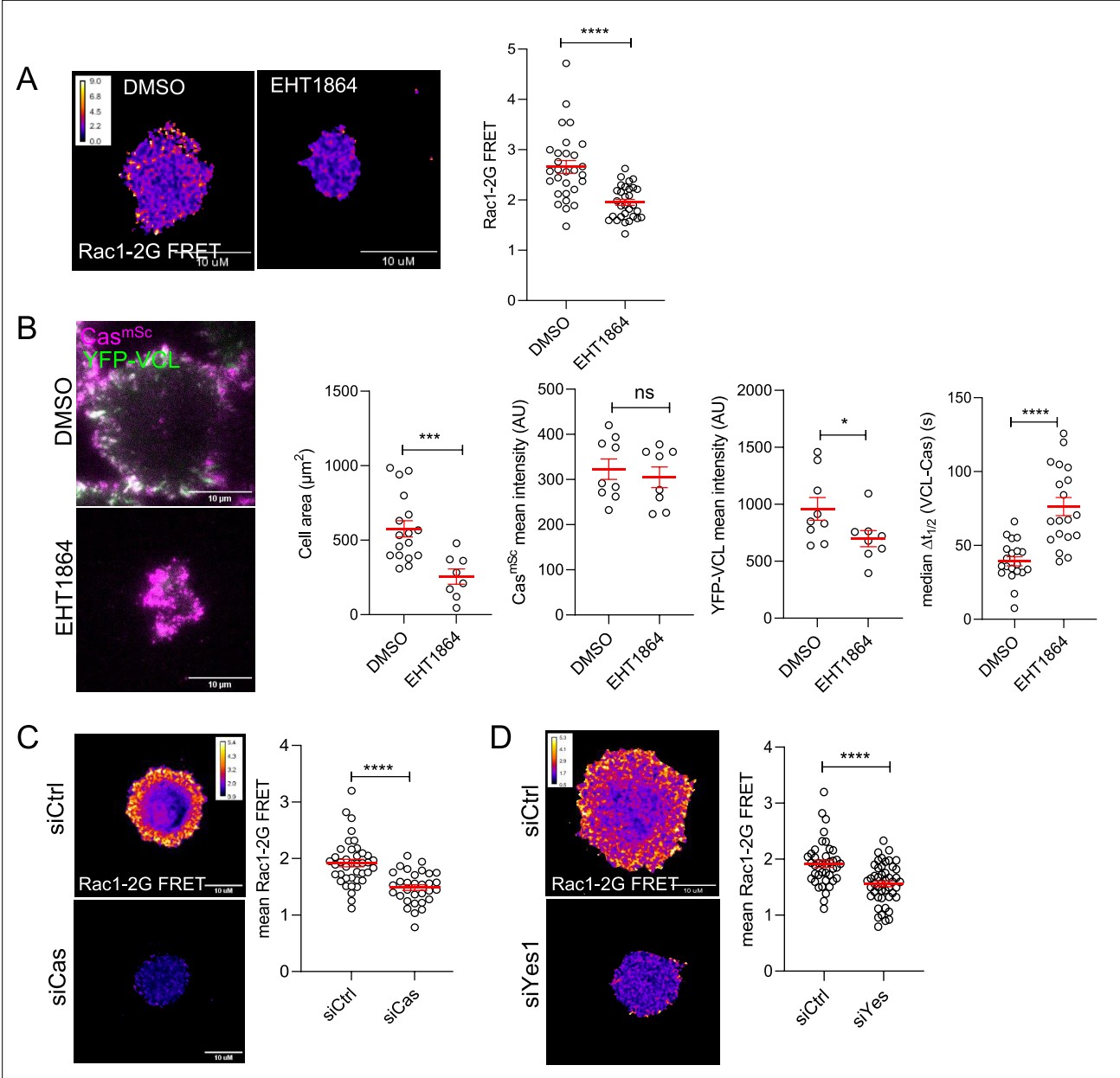

**Figure 7.** Rac1 mediates Cas-dependent cell spreading and adhesion assembly. (**A**) Allosteric inhibitor EHT1864 inhibits Rac1 activation during cell attachment. FRET ratio images (left) and quantification (right) of Rac1-2G MCF10A cells treated with DMSO or EHT1864 and imaged after 30 min attachment. (**B**) Cell spreading and vinculin but not Cas recruitment requires Rac1. Images and quantification of spreading Cas[mSc] YFP-VCL MCF10A cells treated with DMSO or EHT1864. Error bars show mean and standard error of the mean (SEM) of 8–20 cells in three biological repeats. (**C, D**) Rac1 activation requires Cas and Yes1. FRET ratio images (left) and quantification (right) of Rac1-2G MCF10A cells treated with (**C**) control or Cas, or (**D**) control or Yes1 siRNA. Error bars show mean and SEM from >30 cells from three biological replicates. ns, not significant; *p < 0.05; ***p < 0.001; ****p < 0.0001 by Mann–Whitney *U*-tests.

Together, these results suggest a model in which nascent clusters of inactive integrin β1, phosphorylated Cas, active SFKs and Crk stimulate Rac1 and generate ROS, creating positive feedback that strengthens and maintains signaling, fine-tuned by negative feedback from CRL5[SOCS6]. Subsequently, integrin β1 is activated and talin1, kindlin2, vinculin, and other mechanosensing proteins assemble to form a focal adhesion (*Figure 8E*).

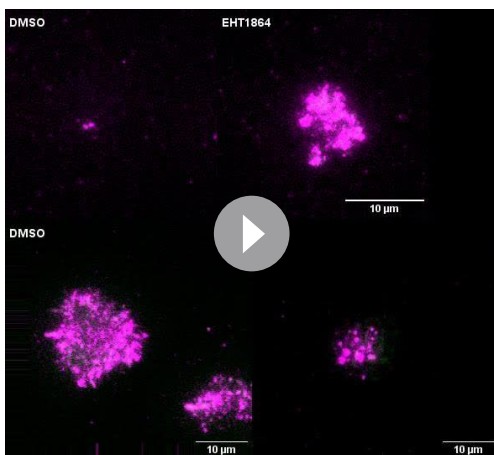

**Video 9.** Rac1 and reactive oxygen species (ROS) requirement. Cas^mSc YFP-VCL MCF10A cells treated with DMSO (top left) or EHT1864 (top right) or with DMSO (bottom left) or diphenylamineiodonium (DPI) (bottom right). 20 s time intervals.

https://elifesciences.org/articles/90234/figures#video9

## Discussion

Recent studies of focal adhesion assembly have elucidated the fundamental role of mechanosensitive proteins such as talin in activating integrins and recruiting vinculin under RhoA-dependent actomyosin tension (*Henning Stumpf et al., 2020*; *Kanchanawong et al., 2010*; *Legerstee and Houtsmuller, 2021*; *Wolfenson et al., 2019*; *Zhu et al., 2021*). Other structural components of focal adhesions are recruited within a few seconds of talin binding and ECM engagement (*Bachir et al., 2014*; *Choi et al., 2008*; *Han et al., 2021*; *Laukaitis et al., 2001*). Steps preceding mechanosensing are unclear, however. Our results suggest a two-step model in which SFKs, Cas, Crk/CrkL, and Rac1 play a key role before talin and vinculin (*Figure 8E*). In the first step, phosphorylated Cas and Crk/CrkL cluster with inactive integrin β1. This step does not require talin1, kindlin2, vinculin, or other structural proteins tested, and these proteins are only present at low level. Moreover, the integrin is inactive, so presumably not bound to the ECM. The clusters appear to grow through positive feedback involving SFKs, Cas, Crk/CrkL, Rac1, and ROS, opposed by negative feedback through CRL5^SOCS6. Clusters may need to reach a critical size for force transmission before the next step can occur (*Coyer et al., 2012*). In the second step, integrin β1 is activated and mechanosensing and structural proteins accumulate. FAK is present at low level in step one but is not activated until step 2, consistent with FAK as a mechanosensor (*Le Coq et al., 2022*). Remarkably, levels of total and phosphorylated Cas decline during the second step. Thus, in our model, Cas regulates the first step of integrin clustering but may not be needed for the final assembly.

Our model is based largely on experiments using epithelial cells attaching to collagen through integrin α2β1 and contrast with previous studies reporting normal focal adhesion assembly when Cas and SFK-mutant fibroblasts were spreading or migrating on FN (*Bockholt and Burridge, 1995*; *Webb et al., 2004*). FN adhesions contain αv as well as β1 integrin heterodimers, so in principle a non-β1 integrin could bypass the need for Cas. Different integrins are known to have different properties. For example, integrin αv adhesions are larger and more reliant on RhoA than integrin β1 adhesions (*Coyer et al., 2012*), and talin1 binds more strongly to β3 than β1 (*Anthis et al., 2010*; *Lu et al., 2016*). However, our preliminary experiments suggest that the two-step model applies also to fibroblasts on FN. We found that Cas is required for epithelial and fibroblast adhesion and spreading on FN, mediated by α5β1 and one or more αv integrins. Cas clusters were also precursors of focal adhesions when epithelial cells spread on FN. A Cas requirement for adhesion assembly in previous studies of mutant fibroblasts may have been hidden by expression of Cas family members or compensation during isolation of the cell lines.

The type of integrin may explain the unexpected finding that Yes1 was limiting for integrin β1 adhesion assembly on collagen. Previous studies showed that Src regulates αvβ3 adhesions but not α5β1 adhesions (*Felsenfeld et al., 1999*), while Fyn and Cas were both needed for force-sensitive integrin αvβ3 adhesions (*Kostic and Sheetz, 2006*; *von Wichert, 2003*). Yes1 has a higher affinity than Src or Fyn for integrin β1 cytoplasmic tails (*Arias-Salgado et al., 2003*; *Arias-Salgado et al., 2005*). In addition, liquid-ordered membrane microdomains (lipid rafts) contain Yes1 and Fyn but not Src (*Resh, 1999*). These microdomains play a poorly understood role in integrin clustering (*Lietha and Izard, 2020*). Therefore, Yes1 preference for membrane microdomains and integrin tails may explain its special role in Cas-dependent integrin clustering.

SFK–Cas–Crk/CrkL–Rac1 signaling not only initiates focal adhesion formation but also cell spreading. Curiously, cell spreading induced by active Rac1 was also Cas dependent. This appears to be due to positive feedback by Rac1-dependent ROS generation and ROS activation of SFK–Cas signaling

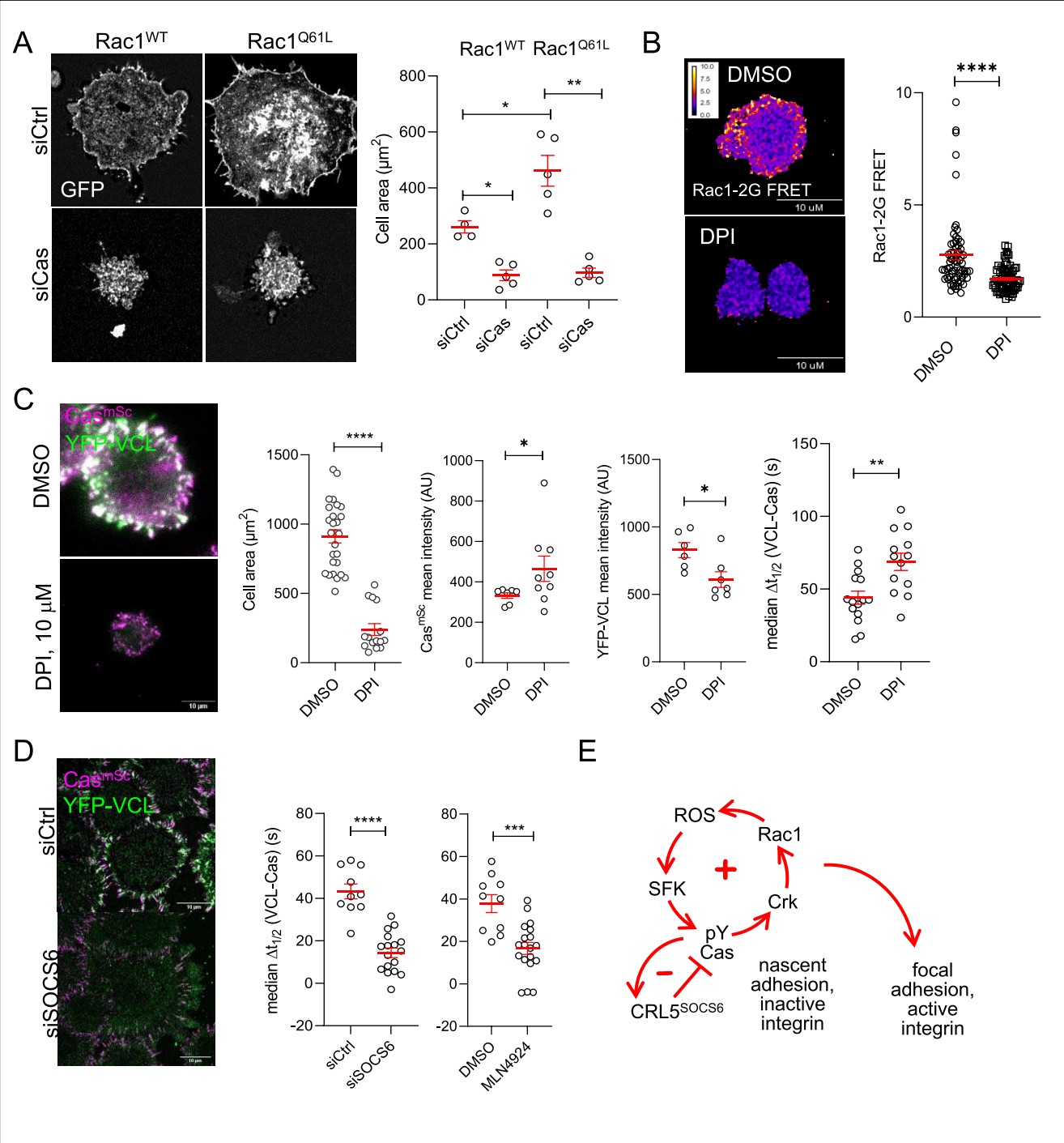

**Figure 8.** Positive and negative feedback regulates focal adhesion assembly. (**A**) Rac1 requires Cas to induce cell spreading. Images and quantification of MCF10A cells expressing EGFP-Rac1WT or -Rac1Q61L that were treated with control or Cas siRNA and fixed after 30 min of spreading. (**B, C**) Reactive oxygen species (ROS) regulates Rac1 activation, Rac1 activation, cell spreading and vinculin recruitment to Cas clusters. (**B**) FRET images and quantification of Rac1-2G MCF10A cells treated with DMSO or NADPH-dependent oxidase inhibitor diphenylamineiodonium (DPI) and fixed after 30 min of spreading. (**C**) Images and quantification of Cas$^{mSc}$ YFP-VCL MCF10A cells treated with DMSO or DPI. Graphs show mean cell area, mean Cas$^{mSc}$ and YFP-VCL intensity, and median $\Delta t_{1/2}$ (VCL-Cas) from 7 to 12 cells in three biological repeats. (**D**) Inhibiting Cullins accelerates vinculin recruitment to Cas clusters. Images and quantification of Cas$^{mSc}$ YFP-VCL MCF10A cells treated with control or SOCS6 siRNA or with DMSO or Cullin Neddylation inhibitor MLN4924. Graphs show median $\Delta t_{1/2}$ (VCL-Cas) from 8 to 12 cells in three biological repeats. All error bars represent mean and standard error of the mean (SEM) and all p values by Mann–Whitney *U*-tests. (**E**) Two-step model. In the first step, co-clustering of Cas with inactive integrin leads to SFK-dependent phosphorylation of Cas, recruitment of Crk/CrkL, activation of Rac1, ROS production, and positive feedback that

*Figure 8 continued on next page*

*Figure 8 continued*

strengthens and maintains signaling to form a nascent adhesion. Positive feedback is opposed by negative feedback resulting from CRL5^SOCS6. In a second step, integrin β1 is activated and talin1, kindlin2, vinculin, actin, and other proteins are recruited to form a focal adhesion. The second step may be triggered by growth of the nascent adhesion to a critical size, or by decreased occupancy with Cas. See Discussion for details.

(*Garton et al., 1996*; *Giannoni et al., 2005*; *Nimnual et al., 2003*; *Tonks, 2005*; *Ushio-Fukai, 2006*). Such feedback may serve to amplify and spread the signal, allowing the initial integrin–Cas clusters to grow in the face of negative feedback through CRL5^SOCS6. However, the positive feedback makes it challenging to identify the exact mechanism for initial cluster growth and later adhesion assembly. For example, SFK–Cas–Crk/CrkL signaling may only be required to activate Rac1, and Rac1 may then induce integrin clustering. Indeed, seminal studies showed that transient expression of active Rac1 induces focal complexes, although the Rac1 effector involved is unclear (*Nobes and Hall, 1995*). That study also showed that Rac1-induced focal complexes only mature into focal adhesions when Rac1 is inhibited, RhoA is activated, and actomyosin tension develops. The loss of Cas from clusters may thus be important in our system to locally decrease Rac1 activity and allow force generation for focal adhesion maturation. Unfortunately, the Rac1 sensor we used did not allow high resolution temporal imaging of Rac1 activity at the level of individual adhesions. Alternatively, Rac1 may only be needed to generate ROS and stimulate Cas phosphorylation, and the latter may induce integrin clustering. This could occur by formation of networks of phospho-Cas, Crk, and multivalent Crk-binding proteins like DOCK180, linked somehow to integrin tails. Recent studies elegantly showed that phospho-Cas forms protein condensates when mixed with an SH2–SH3 protein, Nck, and an Nck-binding protein, N-WASP, in vitro (*Case et al., 2022*). These protein condensates synergize with FAK–paxillin condensates and kindlin to cluster integrin tails on planar lipid bilayers in vitro. It is possible that similar protein condensates or networks form in our system. These condensates could then grow in space and time through Rac1-induced actin polymerization, and positive feedback through ROS.

Our studies also raise the question of how phospho-Cas induces integrin β1 clustering. Cas localization to focal adhesions requires its SH3 and FAT domains (*Donato et al., 2010*; *Nakamoto et al., 1997*). These domains are thought to localize Cas in adhesions by direct binding to FAK, vinculin and paxillin (*Janoštiak et al., 2014a*; *Polte and Hanks, 1995*; *Zhang et al., 2017*). However, FAK, vinculin, and paxillin levels were low in initial phospho-Cas–Crk–integrin clusters and increase when Cas levels decrease, suggesting that Cas associates with integrins through a different mechanism. For example, the integrin β1 tail may bind to Yes1 that is associated with Cas (*Arias-Salgado et al., 2005*), or membrane microdomains could mediate co-clustering of Yes1 with integrins (*Lietha and Izard, 2020*). Whatever the mechanism for cluster formation, cluster growth could then activate associated SFKs by increasing transphosphorylation (*Arias-Salgado et al., 2003*; *Berrier et al., 2002*; *Bodeau et al., 2001*; *Buensuceso et al., 2003*). Further understanding of the precise mechanism will require additional in vivo and in vitro analysis.

## Materials and methods
### Plasmids

pMSCV-Puro-EYFP-vinculin, pMSCV-Puro-EYFP-mCasWT, pMSCV-Puro-EYFP-mCas15F, and pBabe-Puro-mCherry-mVCL were described previously (*Teckchandani and Cooper, 2016*). EGFP-Rac1^WT and EGFP-Rac1^Q61L were provided by K. Wennerberg, University of North Carolina, Chapel Hill, NC (*Arthur et al., 2004*). The following vectors were gifts from the indicated investigators: pmScarlet-i_C1 (Dorus Gadella, Addgene plasmid # 85044), pORANGE cloning template vector (Harold MacGillavry, Addgene # 131471), pCE-mp53DD (Shinya Yamanaka, Addgene # 41856), pLenti-Rac1-2G (Olivier Pertz, Addgene # 66111), pcDNA3.1-mGreenLantern

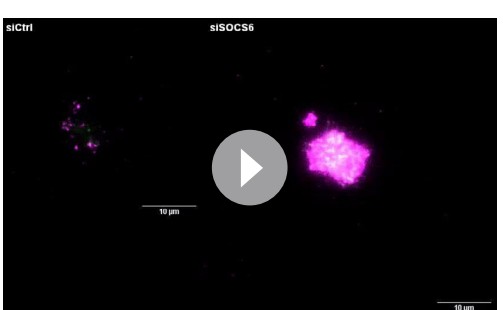

**Video 10.** Stimulation of vinculin recruitment and spreading in SOCS6-depleted cells. Cas^mSc YFP-VCL MCF10A cells treated with control (left) or SOCS6 (right) siRNA. 15 s time intervals.
https://elifesciences.org/articles/90234/figures#video10

(Gregory Petsko, Addgene # 161912) (*Campbell et al., 2020*), pMD2.G and psPAX2 (Didier Trono, Addgene #12259 and 12260). pLenti Ecto-pHluorin β1 integrin with 4-residue linkers was kindly provided by David A. Calderwood (Yale University School of Medicine, USA) (*Huet-Calderwood et al., 2017*).

## Gene editing

We inserted fluorescent protein tags into endogenous genes by homology-independent intron targeting, with modifications (*Serebrenik et al., 2019*; *Zhong et al., 2021*). Intron targeting has several advantages over exon targeting. First, the exact position of artificial exon within the intron is unimportant, allowing insertion at the single guide RNA (sgRNA) SpCas9 target site with highest predicted efficiency. Second, the exon does not need to be inserted precisely, since errors should be corrected by RNA splicing. Thus, homology arms are unnecessary, shortening the donor sequence. Third, by inserting a fluorescent protein open-reading frame lacking an initiation codon, the majority of fluorescent cells should be correctly targeted and can be isolated by FACS without need for single-cell cloning.

We used the pORANGE vector which encodes an sgRNA, SpCas9, and a polylinker for subcloning the donor sequence (*Willems et al., 2020*). Target sites were identified in introns of interest using the CHOPCHOP sgRNA designer (*Labun et al., 2019*). Corresponding sgRNA sequences were synthesized (Integrated DNA Technologies), annealed and inserted into pORANGE using combined restriction digestion and ligation at the BbsI sites. Donor sequences were then inserted at the HindIII (5′) and XhoI or BamHI at (3′) sites using HindIII-HF and XhoI-HF or BamHI-HF (New England Biolabs).

For N-terminal tagging of Cas with mScarlet, the donor contained: (1) a canonical splice acceptor (SA) sequence *Smith, 1997*; (2) mScarlet, lacking its initiation codon and followed by a 24-nucleotide sequence encoding GGMDELYK; (3) a canonical splice donor (SD) sequence (*Connelly and Manley, 1988*). The donor sequence for mScarlet and linker was PCR amplified from pmScarlet-i_C1 using Q5 High-Fidelity DNA polymerase (New England Biolabs).

For C-terminal tagging of ITGB1 and Crk, the donor contained: (1) the SA sequence; (2) the last exon coding sequence with the stop codon replaced by a linker (GGGGARRRGQAGDPPVAT for ITGB1 [*Parsons et al., 2008*] or GGGS for Crk), the fluorescent tag and stop codon; (3) the SV40 3′ processing and polyadenylation sequence (*Connelly and Manley, 1988*). The ends of donor were sandwiched between inverted sites for the gene targeting sgRNA, ensuring donor excision by SpCas9 (*Connelly and Manley, 1988*; *Willems et al., 2020*) (see figure supplements for Cas9 target site orientation). The donor sequences were ordered as gBlocks (Integrated DNA Technologies).

The targeting vectors were transiently transfected with Lipofectamine 2000 into MCF10A epithelial cells or HeLa cells together with pCE-mp53DD, an episomal plasmid encoding a dominant-negative mutant of TP53, to avoid apoptosis due to DNA damage responses (*Haapaniemi et al., 2018*; *Ihry et al., 2018*). Two weeks after transfection, fluorescent cells were checked visually on a Leica Stellaris 5 confocal microscope and selected by FACS.

## Oligonucleotides for *Cas* gene editing

| Name | Sequence and notes |
|---|---|
| **Cas sgRNA** | BbsI-**TARGET** |
| Cas gRNA Fw | caccg**ATCAGCGGTGTTCACTCAAG** |
| Cas gRNA Rv | aaac**CTTGAGTGAACACCGCTGAT**c |
| **Cas mScarlet PCR** | HindIII-TARGET-<u>PAM</u>-*SPLICE ACCEPTOR*-mScarlet |
| mSc Donor HindIII Fw | ataaagctt*ATCAGCGGTGTTCACTCAA* G<u>GGG</u>*CTAATCTCCTCTCTTCTCCTCTCT CCAG*gtgagcaagggcgaggcagt |
| | XhoI-<u>PAM</u>-TARGET-*SPLICE DONOR*-mScarlet_linker |
| mSc Donor XhoI Rv | atactcgag<u>CCC</u>CTTGAGTGAACACCGCT GAT*AACCAATACTTAC*cttgtacagctcgtccatgcc |

*Continued on next page*

*Continued*

| Name | Sequence and notes |
|---|---|
| **Cas Genomic PCR** | |
| a | CACCTCTACATTCTAGCCTGGG |
| b | GAACCTGCAACCCAAAACAC |
| c | GCCCCGTAATGCAGAAGAAG |
| d | GCATGAACTCCTTGATCACTGC |
| **Cas cDNA PCR** | |
| a | TCGGAGCCCCGAGGGCACGCG |
| b | CACGATGCCCTGGCGCCCATG |
| c | CCGCGGCACCAACTTCCCTCC |
| d | CGGGGATGTCGGCGGGGTGCT |

## Oligonucleotides and gBlocks for *Crk* gene editing

| Name | Sequence and notes |
|---|---|
| **Crk sgRNA** | BbsI-**TARGET** |
| Crk gRNA Fw | CACCG**CCCTGCGGCTGGACTTACGT** |
| Crk gRNA Rv | AAAC**ACGTAAGTCCAGCCGCAGGG**C |
| **Crk Genomic PCR** | |
| a | TGACCCATACAGTGACTTCAGG |
| b | TTATGCATCTGGGCTTGTACTG |
| c | GAGCAAAGACCCCAACGAGAA |
| d | GCTGAACTTGTGGCCGTTTAC |
| **Crk cDNA PCR** | |
| a' | CTGATTGGAGGTAACCAGGAG |
| b' | GCAGATGAACTTCAGGGTCAG |
| Crk<sup>mGL</sup> gBlock (Donor) | HindIII-<u>PAM</u>-TARGET-*SPLICE ACCEPTOR*-<u>3'ORF</u>-*GGGS*-*mGL*-*stop*-<u>spacer</u>-SV40polyA-<u>PAM</u>-TARGET-BamHI |

*Continued on next page*

*Continued*

| Name | Sequence and notes |
| --- | --- |
| | agcataaagctt<u>CCA</u>ACGTAAGTCCAGCCGCAGGG*CTAATCTCCTCTCTTCTCCTCTCTCCAGG*TCGGTGAGCTGGTAAAGGTTACGAAGATTAATGTGAGTGGTCAGTGGGAAGGGGAGTGTAATGGCAAACGAGGTCACTTCCCATTCACACATGTCCGTCTGCTGGATCAACAGAATCCCGATGAGGACTTCAGC*ggcgctagcatggtgag caagggcgaggagctgttcaccggggtggtgcccatcctggtcgagc tggacggcgacgtaaacggccacaagttcagcgtgtccggcgaggg cgagggcgatgccacctacggcaagctgaccctgaagttcatctgca ccaccggcaagctgcccgtgccctggcccacc ctcgtgaccaccctga cctacggcgtgcagtgcttcagccgctacccc gaccacatgaagcag cacgacttcttcaagtccgccatgcccgaaggctacgtccaggagcg caccatcttcttcaaggacgacggcaactacaagacccgcgccgag gtgaagttcgagggcgacacccctggtgaaccgcatcgagctgaag ggcatcgacttcaaggaggacggcaacatcctggggcacaagct ggagtacaactacaacagccacaacgtctatatcatggccgacaa gcagaagaacggcatcaaggtgaacttcaagatccgccacaacat cgaggacggcagcgtgcagctcgccgaccactaccagcagaacac ccccatcggcgacggccccgtgctgctgcccgacaaccactacctga gcacccagtccgccctgagcaaagacccccaacgagaagcgcgatc acatggtcctgctggagttcgtgaccgccgccgggatcactctcggc atggacgagctgtacaagtaaagcgctccatggccc*AACTTGTT TATTGCAGCTTATAATGGTTACAAATAAAGCAATAGCATCACAAATTTCACAAATAAAGCATTTTTTTCACTGCATTCTAGTTGTGGTTTGTCCAAACTCATCAATGTATCTTATCATGTCTGGATCTCCCA***ACGTAAGTCCAGCCGCAGGG**ggatcctatgca |

## Oligonucleotides and gBlocks for *ITGB1* gene editing

| Name | Sequence and notes |
| --- | --- |
| **ITGB1 sgRNA** | BbsI-**TARGET** |
| ITGB1 gRNA Fw | CACCG**GCGCCTTCTGTTCACGATAA** |
| ITGB1 gRNA Rv | AAAC**TTATCGTGAACAGAAGGCG**C |
| **ITGB1 Genomic PCR** | |
| a | AGTAACTTCCGTAGGAGACCCC |
| b | CATTCTTGAGTCCTTCCTCCAC |
| c | AACGGCATCAAGGTGAACTTC |
| d | GTAGGTCAGGGTGGTCACGAG |
| **ITGB1 cDNA PCR** | |
| a' | GTGTGGTTGCTGGAATTGTTC |
| b' | GTAGGTCAGGGTGGTCACGAG |
| **ITGB1**[GFP] **gBlock (Donor)** | HindIII-<u>PAM</u>-TARGET-*SPLICE ACCEPTOR*-<u>3'ORF</u>-*linker-GFP-stop-*<u>spacer</u>-*SV40polyA*-<u>PAM</u>-TARGET-BamHI |

*Continued on next page*

*Continued*

| Name | Sequence and notes |
|------|--------------------|
| | agcataaagctt<u>CCT</u>TTATCGTGAACAG AAGGCGC*CTAATCTCCTCTCTTCTCC TCTCTCCAG<u>GGGTGAAAATCCTATTTAT AAGAGTGCCGTAACAACTGTGGTCAA TCCGAAGTATGAGGGAAAA</u>ggaggggg ggggcccggaggcgggggaggcggggatc caccggtcgccaccatggtgagcaagggcgagg agctgttcaccggggtggtgcccatcctggtcgag ctggacggcgacgtaaacggccacaagttcagcg tgtccggcgagggcgagggcgatgccacctacgg caagctgaccctgaagttcatctgcaccaccggcaa gctgcccgtgccctggcccaccctcgtgaccaccctga cctacggcgtgcagtgcttcagccgctaccccgacca catgaagcagcacgacttcttcaagtccgccatgccc gaaggctacgtccaggagcgcaccatcttcttcaagg acgacggcaactacaagacccgcgccgaggtgaagtt cgagggcgacaccctggtgaaccgcatcgagctgaag ggcatcgacttcaaggaggacggcaacatcctggggc acaagctggagtacaactacaacagccacaacgtctat atcatggccgacaagcagaagaacggcatcaaggtg aacttcaagatccgccacaacatcgaggacggcagcg tgcagctcgccgaccactaccagcagaacacccccatcg gcgacggccccgtgctgctgcccgacaaccactacctgag cacccagtccgccctgagcaaagaccccaacgagaagcgc gatcacatggtcctgctggagttcgtgaccgccgccgggatc actctcggcatggacgagctgtacaagtaaagcgct*CCAT GGCCCAACTTGTTTATTGCAGCTTATAATGG TTACAAATAAAGCAATAGCATCACAAATTTC ACAAATAAAGCATTTTTTTCACTGCATTCTAG TTGTGGTTTGTCCAAACTCATCAATGTATCTTA TCATGTCTGGATCTCCCT**TATCGTGAACAG AAGGCGC**ggatcctatgca |

## Cell lines, transfection, and infection

MCF10A cells were originally obtained from Dr. J. Brugge (Harvard Medical School) and confirmed by short tandem repeat (STR) profiling. Cells were cultured in Dulbecco's modified Eagle medium, DMEM/F12 growth media (Thermo Fisher Scientific) supplemented with 5% horse serum (Thermo Fisher Scientific), 10 µg/ml insulin (Thermo Fisher Scientific), 0.1 µg/ml cholera toxin (EMD Millipore), 0.5 µg/ml hydrocortisone (Sigma-Aldrich), and 20 ng/ml epidermal growth factor (EGF) (Thermo Fisher Scientific), and passaged using trypsin/ethylenediaminetetraacetic acid (EDTA) or Accutase (Sigma-Aldrich A6964). For experiments, cells were detached with Accutase and resuspended in assay media (DMEM/F12, 2% horse serum, 0.1 µg/ml cholera toxin, 10 µg/ml insulin, 0.5 µg/ml hydrocortisone, and 0 ng/ml EGF).

HeLa (RRID:CVCL_0030) cells were initially obtained from ATCC and confirmed by STR DNA profiling. Cells were cultured in DMEM supplemented with 10% fetal bovine serum (FBS) and penicillin/streptomycin (both 100 U/ml) and passaged with trypsin/EDTA. For experiments, cells were detached using Accutase and resuspended in DMEM without serum.

HFFs were originally from Dr. Denise Galloway, Fred Hutchinson Cancer Center (*Passalaris et al., 1999*). They were cultured in DMEM supplemented with 10% FBS, 100× nonessential amino acids (NEAA) and penicillin/streptomycin (both 100 U/ml) and passaged using Accutase. For experiments, cells were detached using Accutase and resuspended in DMEM supplemented with NEAA without serum.

Retro- and lentiviruses were generated by transfecting 293FT cells with viral vector, pMD2.G and psPAX2 in 2:1:2 ratio with PolyJet transfection reagent (SignaGen Laboratories). Media were harvested 2 days later and added to recipient cells with 1 µg/ml polybrene (Sigma) for 8–16 hr. Expressing cells were checked visually using a Leica Stellaris 5 confocal microscope and selected using 1 µg/ml puromycin or FACS, depending on the vector.

## Antibodies

Following antibodies were used: mouse anti-Cas (610271) (BD Biosciences); rabbit-phospho-Y410Cas (4011S) and phospho-Y249Cas (4014S) (Cell Signaling Technology); mouse anti-vinculin (V9131, Sigma-Aldrich); mouse anti-Crk (610035) (BD Transduction labs); mouse anti-CrkL (05-414) (Upstate); sheep anti-paxillin (AF4259, R and D Systems); rabbit anti-pY31 paxillin (44-720G, Biosource); rat anti-integrin-β1 (9EG7, 553715) and mouse anti-paxillin (610051) (BD Biosciences); rabbit anti-Talin1 (A14168-1-AP), rabbit-anti Kindlin2 (11453-1-AP), and mouse anti-FAK (66258-1-Ig) (Proteintech); rat anti-integrin-β1 (mAB13, MABT821), rabbit anti-integrin β1 (AB1952P), mouse anti-integrin β1 (12G10, MAB2247) (Millipore); rabbit-anti-integrin-β3 (A2542), rabbit-anti-integrin-αv (A2091) (abclonal); rat anti-integrin-β1 (AIIB2), mouse anti-integrin α5β1 (P8D4), and mouse anti-integrin α2β1 (P1E6) (Developmental Studies Hybridoma Bank); AlexaFluor 488 goat anti-rabbit IgG (H+L), AlexaFluor 488 goat anti-mouse IgG (H+L), AlexaFluor 647 goat anti-mouse IgG (H+L), AlexaFluor 647 goat anti-sheep IgG (H+L), and AlexaFluor 633 goat anti-rat IgG (H+L) (Invitrogen); IRDye 680RD goat anti-mouse and IRDye 800CW goat anti-rabbit (LI-COR).

## Inhibitors

| Inhibitor | Source | Concentration |
|---|---|---|
| DPI | EMD Millipore, Cat: 300260-10MG | 10 µM |
| eCF506 | Cayman Chemical, Cat: 19959 | 100 nM |
| EHT1864 | ApexBio, Cat: B5487 | 10 µM |
| MLN4924 | Fisher, Cat: 50161353 | 5 µM |

## siRNA transfection

Cells were suspended in growth media and added to dishes with 50 pmol pooled siRNA and RNAiMAX (Invitrogen) as per the manufacturer's instructions. Transfection was repeated 2 days later and cells analyzed after a further 2 days.

| Negative control siRNA | QIAGEN, Cat: 1027280 | AAT TCT CCG AAC GTG TCA CGT |
|---|---|---|
| siFyn | Dharmacon, Cat: L-003140-00-0005 | J-003140-11: CGG AUU GGC CCG AUU GAU A<br>J-003140-12: GGA CUC AUA UGC AAG AUU G<br>J-003140-13: GAA GCC CGC UCC UUG ACA A<br>J-003140-14: GGA GAG ACA GGU UAC AUU C |
| siCrk | Dharmacon, Cat: M-010503-03-0005 | D-010503-02: GGA GAC AUC UUG AGA AUC C<br>D-010503-03: UCC CUU ACG UCG AGA AGU A<br>D-010503-04: GGA CAG CGA AGG CAA GAG A<br>D-010503-19: GGG ACU AUG UGC UCA GCG U |
| siCrkL | Dharmacon, Cat: M-012023-02-0005 | D-012023-01: CCG AAG ACC UGC CCU UUA A<br>D-012023-02: GAA GAU AAC CUG GAA UAU G<br>D-012023-05: AAU AGG AAU UCC AAC AGU U<br>D-012023-18: AGU AAA ACU UAA CGG ACU U |
| siSOCS6 | QIAGEN, Cat: GS9306 | SI03068359: CAG CTG CGA TAT CAA CGG TGA<br>SI00061383: TAG AAT CGT GAA TTG ACA TAA<br>SI00061376: CGG GTA CAA ATT GGC ATA ACA<br>SI00061369: TTG ATC TAA TTG AGC ATT CAA |
| siYes1 | QIAGEN, Cat: GS7525 | SI02223942: GAG GCT CCT GCT TAT TTA TAA<br>SI02223935: CCA GCC TAC ATT CAC TTC TAA<br>SI00302218: AAT CCC TCC ATG AAT TGA TGA<br>SI02635206: AAG TAT AAT GCA GTA CAT TAA |
| siBCAR1 (Cas) | QIAGEN, Cat: GS9564 | SI02757741: AAG CAG TTT GAA CGA CTG GAA<br>SI02757734: CTG GAT GGA GGA CTA TGA CTA<br>SI04438280: CCA GGA ATC TGT ATA TAT TTA<br>SI04438273: CAA CCT GAC CAC ACT GAC CAA |

*Continued on next page*

*Continued*

| | | |
|---|---|---|
| **Negative control siRNA** | **QIAGEN, Cat: 1027280** | **AAT TCT CCG AAC GTG TCA CGT** |
| Human-specific siBCAR1 (Cas) | QIAGEN | SI00106876: TTGACTAAGAGTCTCCATTTA<br>SI03065874: CAGCATCACGCGGCAGGGCAA<br>SI04438273: CAACCTGACCACACTGACCAA<br>SI04438280: CCAGGAATCTGTATATATTTA |
| siSrc | QIAGEN, Cat: GS6714 | SI02664151: CTC CAT GTG CGT CCA TAT TTA<br>SI02223928: CGG CTT GTG GGT GAT GTT TGA<br>SI02223921: AAG CAG TGC CTG CCT ATC AAA<br>SI03041605: ACG GCG CGG CAA GGT GCC AAA |
| siVinculin | Dharmacon, Cat: L-009288-00-0005 | J-009288-05: UGA GAU AAU UCG UGU GUU A<br>J-009288-06: GAG CGA AUC CCA ACC AUA A<br>J-009288-07: GCC AAG CAG UGC ACA GAU A<br>J-009288-08: CAG CAU UUA UUA AGG UUG A |
| siPTK2 (FAK) | Dharmacon, Cat: L-003164-00-0005 | J-003164-13: GCG AUU AUA UGU UAG AGA U<br>J-003164-14: GGG CAU CAU UCA GAA GAU A<br>J-003164-15: UAG UAC AGC UCU UGC AUA U<br>J-003164-16: GGA CAU UAU UGG CCA CUG U |
| siPaxillin | Dharmacon, Cat: L-005163-00-005 | J-005163-05: CAA CUG GAA ACC ACA CAU A<br>J-005163-06: GGA CGU GGC ACC CUG AAC A<br>J-005163-07: CCA AAC GGC CUG UGU UCU U<br>J-005163-08: UGA CGA AAG AGA AGC CUA A |
| siFERMT2 (Kindlin2) | Dharmacon, Cat: L-012753-00-0005 | J-012753-05: GCC CAG GAC UGU AUA GUA A<br>J-012753-06: CUA CAU AUU UCU CUC AAC A<br>J-012753-07: GAA CUG AGU GUC CAU GUG A<br>J-012753-08: AAU GAA AUC UGG CUU CGU U |
| siTalin1 | Dharmacon, Cat: L-012949-00-005 | J-012949-05: GAA GAU GGU UGG CGG CAU U<br>J-012949-06: GUA GAG GAC CUG ACA ACA A<br>J-012949-07: UCA AUC AGC UCA UCA CUA U<br>J-012949-08: GAG AUG AGG AGU CUA CUA U |
| siITGB1 | Santa Cruz, Cat: sc-35674 | sc-35674A: GAGAUGAGGUUCAAUUUGATT<br>sc-35674B: GAUGAGGUUCAAUUUGAAATT<br>sc-35674C: GUACAGAUCCGAAGUUUCATT |
| siITGAV | Santa Cruz, Cat: sc-29373 | sc-29373A: GCAUCUAUCUUGAAAGUAATT<br>sc-29373B: CUGGUUUGAACGAUAGAAATT<br>sc-29373C: GAAGCUGUGUAGUAUAUCATT |

## Cell lysis and immunoblotting

Cells were harvested after 30 min of attachment. Cells were washed three times with cold phosphate-buffered saline (PBS) followed by lysis in radioimmunoprecipitation assay (RIPA) buffer (1% Triton X-100, 1% sodium deoxycholate, 0.1% sodium dodecyl sulfate [SDS], 20 mM Tris–HCl pH 7.4, 150 mM NaCl, 5 mM ethylene glycol tetraacetic acid (EGTA)) with freshly added protease and phosphatase inhibitors (10 µg/ml Aprotinin, 1 mM phenylmethylsulfonyl fluoride (PMSF), 1 mM sodium vanadate) on ice. The lysates were collected after 30 min of incubation on ice and centrifuged at 12,000 rpm for 10 min at 4°C. Supernatants were collected and adjusted to equal protein concentration using the Pierce BCA protein assay kit.

Lysates were adjusted to SDS sample buffer, heated at 95°C, and resolved on SDS–polyacrylamide gel electrophoresis (PAGE) using 15% polyacrylamide/0.133% bis-acrylamide or 12.5% acrylamide/0.1% bis-acrylamide gels, and transferred on to nitrocellulose membrane. Blocking was performed in Odyssey blocking buffer (LI-COR Biosciences) supplemented with 5% bovine serum albumin (BSA) for 30 min. After blocking, membrane was probed with the primary antibody overnight, washed in Tris-buffered saline 0.1% Tween 20, followed by incubation for 45 min with IRDye 800CW goat anti-rabbit or 680RD goat anti-mouse-conjugated secondary antibodies. Images were collected using the Odyssey Infrared Imaging System (LI-COR Biosciences) and quantified using ImageJ.

## Cell spreading and migration assays

Cells were treated with siRNA using the double transfection method as described above. Cells were starved overnight in assay media (MCF10A) or DMEM (HeLa), then detached with Accutase and resuspended in assay media or DMEM. Cells were incubated for 60 min in 5% $CO_2$ at 37°C in suspension before adding to glass-bottom dishes (FluoroDish, FD35-100, World Precision Instruments) or 12 mm diameter coverslips (Fisherbrand 1254580) that had been previously coated with 50 µg/ml collagen-I (Advance Biomatrix, #5056) or 5 µg/ml FN (Sigma, #F1141) for at least 3 hr at 37°C and washed with PBS. Pharmacological agents were added just before seeding cells for imaging.

## Live imaging

Dual-color imaging of live cells was performed for a 30-min time period, either immediately after plating the cells, to record attachment and spreading, or approximately 24 hr after plating, to record spontaneous lamellipodia formation. Images were recorded in 4–5 fields of view on a fully automated TIRF microscope (Nikon Ti, ×100/1.49 CFI Apo TIRF oil immersion objective) equipped with Perfect Focus, motorized x–y stage, fast piezo z stage, stage-top incubator with temperature and $CO_2$ control, and Andor iXon X3 EMCCD camera with 512 × 512-pixel chip (16 µm pixels). The images were acquired using Nikon NIS Elements software and processed using ImageJ. Frame rate varied between 15 and 20 s depending on the number of fields of view recorded. Slight drift in the dual color time-lapse images was corrected using ImageJ registration tool Image Stabilizer.

Cell migration was imaged once every 15 min for 12 hr using phase-contrast microscopy on an IncuCyte S3 and analyzed using ImageJ.

## Immunofluorescence imaging

MCF10A Cas$^{mSc}$ cells, either expressing YFP-vinculin or not, were plated on glass-bottom dishes or coverslips and allowed to attach for 30 min before fixation with 4% paraformaldehyde in PBS for 15 min. Cells were permeabilized with 0.1% Triton X-100 in PBS for 5 min, and blocked for an hour in 5% normal goat serum, 2% BSA in PBS, all at room temperature. The cells were incubated with different combinations of primary antibodies (1:200 dilution) for either 2–3 hr at room temperature or overnight at 4°C, then washed gently three times. Alexa Fluor-conjugated secondary antibodies were added at 1:200 dilution for 1 hr at room temperature. Alexa Fluor 488- or 647-conjugated phalloidin was used for F-actin. Glass-bottom dishes were left in PBS for TIRF microscopy as above. Coverslips were mounted in ProLong Gold (Invitrogen) for confocal microscopy using a Dragonfly 200 High-speed Spinning disk confocal imaging platform (Andor Technology Ltd) on a Leica DMi8 microscope stand equipped with a ×100/1.4 oil immersion objective, iXon EMCCD and sCMOS Zyla cameras and Fusion Version 2.3.0.36 (Oxford Instruments) software together with Imaris simultaneous deconvolution. The most ventral plane was used for quantification.

## Immunofluorescence quantification

Macros were written in ImageJ for uniform image processing and analysis. Backgrounds were subtracted and intensities adjusted equally across each image set.

For analysis of the spatial distribution of the proteins and phosphoproteins within Cas/vinculin clusters, a line (5 × 25-pixels width × length, 0.16 µm/pixel) was plotted along the major axis of a Cas$^{mSc}$ cluster, starting from the outer edge and moving inwards. The intensity profile for each antibody along this line was quantified in ImageJ and aligned using Cas$^{mSc}$ as a reference point. The mean normalized intensity profile for each antibody was then calculated across 20–25 regions from several cells. A heat map for the mean intensity profile of the normalized value was generated using GraphPad Prism.

To analyze the number and intensity of clusters containing either Cas or vinculin, or both, images were first spit into individual channels and then summed using the 'Image calculator' command in ImageJ. This summed image was used to generate a binary image mask by applying manual threshold through the Yen method. ROIs >20 pixels (0.52 µm$^2$) were then counted and their mean areas and mean Cas and vinculin intensities quantified. Cell area was quantified either manually by drawing around the cell or from a binary mask created by thresholding through the Triangle method, setting minimum size as 50 pixels.

## Quantification of cluster kinetics from two-color TIRF videos

TIRF image datasets were exported as two-channel time-series hyperstacks in TIF format. Quantification of the time shift between red (Cas$^{mSc}$) and green (YFP-VCL, β1-EctopH, ITGB1$^{GFP}$ or Crk$^{mGL}$)

cluster formation was performed in MATLAB (R2021b). The pipeline involves the following steps: drift correction, image preprocessing and denoising, focal adhesion segmentation and tracking, intensities extraction and normalization, and signal analysis.

Drifts between time frames were corrected by registering the image of one frame to the image of the previous frame using a translation transformation. Drift corrected timeseries were then denoised with a median filter and their background was equalized with a top-hat transform. Clusters were segmented separately in each channel at each time frame by intensity thresholding and size filtering. Threshold values were arbitrarily defined as a quarter of the intensity value of the 50th brightest pixel of the dataset for each channel, and the union of the binary masks of the two channels with an area greater than 20 pixels (0.52 μm$^2$) was used to define clusters. Clusters were tracked over time by creating a 3D stack (XYT) and by computing the resulting connected components. Only clusters tracked in three or more frames (≥40 s) were quantified. For each tracked cluster, the average mean intensity of each channel over time was extracted by first measuring the mean intensity within the mask at each time frame from time zero to the end, resulting in a number of intensity traces equal to the number of time frames where the cluster was tracked.

To calculate the time shift between the red and green channel intensity profiles for each tracked cluster, average mean intensity traces were rescaled to the [0 1] value range, and after smoothing the signal by applying a moving average filter, the time at which the rescaled intensity reaches 0.5 ($t_{1/2}$) was interpolated. The time shift was finally calculated by subtracting red (Cas) $t_{1/2}$ from green (vinculin, integrin, or Crk) $t_{1/2}$.

## Adhesome isolation

Adhesome samples were isolated as described with minor modifications (*Schiefermeier et al., 2014*). MCF10A cells were detached from near-confluent 5 cm plates, resuspended in assay media, incubated in suspension for 30 min, then re-seeded onto an equal number of 5 cm plates that had been precoated with 50 μg/ml collagen. After 1 hr at 37°C, one dish was washed and lysed in 400 μl RIPA buffer to provide a sample of total protein (T) and the other dishes were washed gently with room temperature PBS and incubated with 2 ml freshly diluted 0.5 mM dithiobis(succinimidyl propionate) (EMD Millipore 322133) 0.05 mM (1,4-di [3'-(2'-pyridyldithio)propionamido] butane) (Sigma 16646) in PBS for 5 min at room temperature. Cross-linking was terminated by washing twice with 50 mM Tris 140 mM NaCl pH 7.4 before transferring to ice. 400 μl RIPA buffer was added and the plates were rocked at 4°C for 1 hr. Supernatants (S) were collected in a 2-ml microtube (Axygen MCT-200-L-C). The plates were washed twice with PBS, drained, and 400 μl RIPA buffer added. 2-Mercaptoethanol was added to 2% concentration to all samples, and all microtubes and dishes were sealed and incubated at 50°C for 1 hr. After cooling to room temperature, the dish was scraped and the adhesome (A) fraction was transferred to a 2 ml microtube. Total, supernatant, and adhesome fractions were sonicated (10 s, microtip) to shear DNA and the proteins were precipitated by adding 1.6 ml acetone and placing at −20°C for 1 hr. After centrifugation (10 min, 14,000 rpm), pellets were drained, dried in a stream of air, and incubated with 25 μl 5× concentrated SDS sample buffer at 95°C for 5 min, then 100 μl RIPA buffer was added to all tubes. Samples were centrifuged and equal volumes analyzed by SDS–PAGE, typically on 10% acrylamide/0.13% bisacrylamide gels, followed by Western blotting. The procedure was scaled up as needed to run replicate blots for probing with different antibody combinations. On occasion, dilutions of the T sample were loaded to estimate detection sensitivity.

## Ratiometric FRET imaging

The Rac1-2G reporter was expressed in MCF10A cells by lentiviral transduction. After siRNA treatment, cells were allowed to attach to collagen for 30 min in the absence or presence of various inhibitors, then fixed and mounted. Coverslips were imaged on a Leica SP8 confocal microscope using Leica HCX Plan Apo ×63/1.40 oil immersion objective. Excitation and emission wavelengths as follows: donor (mTFP1) excitation 440 nm, emission 450–510 nm: FRET excitation 440 nm, emission 515–600 nm; acceptor (mVenus) excitation 514 nm, emission 515–600 nm. All channels were collected on the HyD detectors. Images were processed with the Lightening deconvolution (Leica

LASX software) and the FRET ratio in the most ventral plane was quantified using ImageJ as described (*Kardash et al., 2011*).

## Statistics

Data were analyzed using GraphPad Prism. Median and 95% confidence intervals were calculated for non-normal distributions of measurements from single cells. Data from multiple cells or biological replicates were assumed to follow normal distributions allowing calculation of mean and SEM in cases where data from multiple cells in biologically independent experiments were combined. Pairwise comparisons between control and experimental populations testing independent hypotheses were made using the non-parametric Mann–Whitney *U*-test. Experiments testing alternative hypotheses were analyzed using the non-parametric Kruskal–Wallis analysis of variance followed by Dunn's multiple comparison test.

## Acknowledgements

We are very grateful to Lena Schroeder, Peng Guo, and Jin Meng for training and use of equipment in the Cellular Imaging Shared Resource and Luna Yu for computational assistance. We thank Dorus Gadella, Harold MacGillavry, Shinya Yamanaka, Olivier Pertz, Gregory Petsko, and Didier Trono for Addgene plasmids, David Calderwood and Susumu Antoku for additional constructs, and Denise Galloway for cells. Reinhard Faessler provided a detailed protocol for adhesome isolation. We are very grateful for helpful discussions with Matt Berginski and Matthew Kutys. Dayoung Kim, Jihong Bai, Cecilia Moens, Susan Parkhurst, and other colleagues provided useful feedback during development of the project. This research was supported by institutional funds from Fred Hutch and National Institutes of Health grant R01 GM109463 from the US Public Health Service. The Cellular Imaging and Flow Cytometry Shared Resources of the Fred Hutch/University of Washington Cancer Consortium is supported by P30 CA015704.

## Additional information

### Competing interests

Jonathan A Cooper: Senior editor, *eLife*. The other authors declare that no competing interests exist.

### Funding

| Funder | Grant reference number | Author |
|---|---|---|
| Fred Hutchinson Cancer Center | | Saurav Kumar<br>Amanda Stainer<br>Christopher Simpkins<br>Jonathan A Cooper |
| National Institutes of Health | R01 GM109463 | Saurav Kumar<br>Amanda Stainer<br>Christopher Simpkins<br>Jonathan A Cooper |
| National Institutes of Health | P30 CA015704 | Julien Dubrulle |
| Fred Hutch/University of Washington Cancer Consortium | | Jonathan A Cooper |

The funders had no role in study design, data collection, and interpretation, or the decision to submit the work for publication.

### Author contributions

Saurav Kumar, Conceptualization, Data curation, Formal analysis, Validation, Investigation, Visualization, Methodology, Writing - original draft, Writing – review and editing; Amanda Stainer, Data curation, Formal analysis, Validation, Investigation, Visualization, Methodology, Writing - original draft;

Julien Dubrulle, Data curation, Software, Methodology; Christopher Simpkins, Formal analysis, Investigation, Visualization, Methodology, Writing – review and editing; Jonathan A Cooper, Conceptualization, Data curation, Formal analysis, Supervision, Funding acquisition, Validation, Investigation, Visualization, Methodology, Writing - original draft, Writing – review and editing

**Author ORCIDs**
Saurav Kumar http://orcid.org/0000-0002-0992-589X
Amanda Stainer http://orcid.org/0000-0001-7068-2658
Julien Dubrulle http://orcid.org/0000-0002-4186-7749
Christopher Simpkins https://orcid.org/0000-0003-3174-6609
Jonathan A Cooper http://orcid.org/0000-0002-8626-7827

**Decision letter and Author response**
Decision letter https://doi.org/10.7554/eLife.90234.sa1
Author response https://doi.org/10.7554/eLife.90234.sa2

---

# Additional files

**Supplementary files**
• MDAR checklist

**Data availability**
All raw Western blots generated during the study have been included as source files. Matlab code used in the study is uploaded on GitHub (https://github.com/FredHutch/Cas-integrin-paper-2023-Cooper-lab, copy archived at *Fred Hutchinson Cancer Center, 2023*).

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
