## [Editor Report]

This important study advances our understanding of adhesion formation in migrating cells by showing that clustering of the adaptor protein Cas and its binding partners represents the initial step in adhesion formation that occurs before integrin clustering. The evidence supporting the conclusions is convincing overall, although in a few cases, quantifications are based on limited datasets.

---

## [Decision Letter]

**Decision letter after peer review:**

[Editors’ note: the authors submitted for reconsideration following the decision after peer review. What follows is the decision letter after the first round of review.]

Thank you for submitting the paper "Cas phosphorylation regulates focal adhesion assembly" for consideration by *eLife*. We sincerely apologise for the delay in reviewing your manuscript, which was due to the fact that it was difficult to find peer reviewers. Your article has been reviewed by 3 peer reviewers, and the evaluation has been overseen by a Reviewing Editor and a Senior Editor. The following individual involved in the review of your submission has agreed to reveal their identity: Mike Sheetz (Reviewer #3).

Comments to the Authors:

We are sorry to say that, after consultation with the reviewers, we have decided that this work will not be considered further for publication by *eLife*. All three reviewers found that the work is potentially novel and important, and would be interesting to the field. However, very significant concerns were raised. Specifically, reviewers would like to see stronger proof that b1 integrin is the only relevant β integrin in the system you use; they would like to see controls excluding the potential involvement of b6 integrin, as well as experiments in fibroblasts knockout for b3 integrin. There were also concerns about the choice of the integrin antibodies used, the small number of cells analyzed, and some important controls of the tagging strategies missing. Addressing all these concerns is likely possible but will also be laborious and will take more time and effort than can be expected from a normal revision at *eLife*. We, therefore, return the paper to you so that you can send it to another journal if you seek rapid publication. However, we will be prepared to reconsider the manuscript if you can thoroughly address the reviewers' comments. In this case, we will do our best to send the paper to the same reviewers.

*Reviewer #1 (Recommendations for the authors):*

The authors have investigated the role of the p130-cas/crk complex in cell spreading and the hierarchy of adhesion formation in epithelial cells. The authors propose a model whereby phosphorylated Cas (p130Cas, BCAR1), its binding partner, Crk, and inactive FAK and inactive b1-integrins cluster together at the leading edge of the cell protrusion. This they suggest precedes focal adhesions formation with F-actin, active integrins, active FAK, vinculin, and talin. They show convincingly that p130cas depletion severely impairs cell spreading downstream of integrin activation, as integrin activation with Mn^2+^ fails to rescue the spreading defect. The model is attractive; however, it does not seem to be supported fully by the data. The biggest problem is the unfortunate choice of the integrin antibodies used in the paper. Unfortunately, the authors have been misinformed in their choice of the antibodies used (Please see Byron et al., JCS PMID: 19910492). They indicate that they have chosen an antibody that specifically recognizes the active ligand-bound state of integrins. However, the 9EG7 antibody does not recognize the open conformation but it is specific to the extended primed state. Instead, 12G10 for example would recognize this state. They also indicate that they have used an antibody that recognizes the bent inactive state that is not ligand bound. Unfortunately, AIIB2 is not conformation-specific but rather blocks directly the ligand binding to the integrins and thus does not stain inactive integrins. This in fact is obvious from looking at the example stainings provided where their inactive beta1 staining is clearly in adhesions. In most epithelial cells, inactive b1-integrin (detected with Mab13, PIH5, or 4B4 antibodies specifically recognizing inactive integrins) localise primarily to membrane ruffles above the FA staining detected by TIRF.

A big part of the data included in the main figures describes the approaches the authors have used to generate endogenously tagged cell lines for some of the components investigated. This is commendable but as such genomic editing is an established methodology, these details could be moved to the supplements.

Throughout the manuscript, the quantifications are based on rather low numbers of cells, and for some experiments, it is not clear if these are from a single experiment or from biologically independent experiments. In addition, there are several instances where the authors show just a small magnified area of the cells. Smaller magnification images should be provided as well to help the reader to evaluate the quality of the staining overall and how representative the chosen areas are.

Specific points:

1) For endogenous tagging of Cas, the authors employed the insertion of an artificial exon containing mScarlet (mSc) into the first intron of the Cas gene. In the resulting protein, mSc is inserted between Pro4 and Asn5 of Cas without any further linker sequence, thus potentially affecting some of the functions of the Cas SH3 domain. Moreover, phosphorylation of Tyr12 in the Cas SH3 domain has been reported to regulate interaction with several SH3-binding proteins, including vinculin (PMID: 21937722, PMID: 28808245, PMID: 23974298). The authors should provide evidence that inserting mSc in close proximity to the Cas SH3 domain and an important regulation site (Y12) does not affect downstream signalling of the endogenously tagged Cas. For instance, the authors should analyse the interaction of Cas and CasmSc with SH3 domain-binding proteins or analyse Y12 phosphorylation levels of Cas and CasmSc in cells with and without Src expression.

2) In Figure 1B authors show levels of Cas in the CasmSc cells. While most of the Cas in CasmSc cells is tagged, the phosphorylation pattern as shown by pY410 Ab does not replicate the pattern of the total Cas antibody, suggesting phosphorylation of the tagged Cas is substantially weaker than that of the remaining untagged Cas. Authors should provide further evidence that their tagging strategy does not influence Cas substrate domain phosphorylation and thus signalling downstream of the phosphorylated SD (such as interaction with Crk) when compared to untagged Cas.

3) In Figure 2C the authors show data points of 11 cells on Fn and 7 cells on collagen (although the legend indicates n=7-13). Measurement of more cells should be provided in order to strengthen this observation, given the median Δt_1/2_ (VCL^-^Cas) ranges from less than 0s to more than 120s. Moreover, based on this data authors suggest that "Cas clusters may specify localization of integrin α5β1 and α2β1 adhesions", however, this is not obvious from the data provided in this section.

4) Figure 3. Please show the overall distribution of the tagged integrins in the cell. How well do the tagged integrins overlap with the endogenous ones? Please show b1-integrin blots alongside the GFP in the WB.

5) Figure 5. The authors need to provide representative images (lower and higher mag) of all the antibody stainings they have quantified in 5C. How has the specificity of the antibodies/staining been determined? Please see above the comments related to the integrin antibodies used.

6) Figure 6 and 7. How have the authors controlled for off-target effects of their siRNAs? The vinculin silencing efficacy in Figure 6 is poor and precludes making any conclusions on the role of vinculin.

In Figure 6C the authors analyse the role of Cas depletion on the regulation of Mn^2+^-dependent spreading. As a further control, the authors could include a rescue experiment with Cas WT and Cas lacking the CCH domain in Cas-depleted cells.

7) The results in Figure 7A suggest that the interaction of Crk with phosphorylated Cas accelerates the recruitment of Vinculin to the Cas-Crk clusters. Do the authors have a mechanistic explanation of how this could be regulated? Also, to support this data, the authors should perform a Crk rescue experiment in Crk-depleted cells to see whether recruitment of Vinculin to Cas clusters will be rescued to control levels.

8) In Figure 8D, the fluorescence intensity of CasmSc seems to be lower in siSOCS6 than in siCtrl cells. This is rather counterintuitive since the depletion of SOCS6 should stabilize Cas levels. Could the authors comment on this?

Other points:

1) In the introduction, the authors mention the mechanoresponsive roles of talin in adhesion maturation and assembly. It would be beneficial to mention in this context also the effect of the force-mediated extension of the Cas substrate domain on its downstream signalling (PMID: 17129785, reviewed in PMID: 25062607).

2) Figure 1 supplement 2. What is the difference between Hela panels E and F?

3) It is not clear, whether the cells measured in Figure 3 are seeded on Fn or Col-I. If on Fn – would the delay in ITGB1 arrival to Cas clusters be different on Col-I, since authors show there are more 9EG7-positive ITGB1 clusters in cells seeded on collagen (Figure 2B)?

4) This study is in line with previous work (PMID: 30639111), where pY410 Cas has been shown to co-localize with Crk in filopodia tips in a CCH domain-dependent manner and extends these observations to the edge of spreading/migrating MCF10A and HeLa cells. This could perhaps be discussed.

*Reviewer #2 (Recommendations for the authors):*

The manuscript "Cas phosphorylation regulates focal adhesion assembly" by Kumar et al. investigates the role of Cas in cell spreading and finds that Cas, Crk, and FAK cluster together at the cell periphery to initiate focal adhesion formation. In contrast to the previous understanding, they suggest that Cas clustering happens before integrin clustering and hence is the initial step in adhesion formation. This notion is the key novel aspect of the study, however, it is based on the assumption that b1 integrin is the only relevant integrin in the system, which at this point is not supported by the data.

Specific comments:

– The central finding of the study (i.e. that Cas clusters before integrin) is based solely on the study of b1-integrin and hence on the assumptions that it is the only integrin that is relevant for adhesion formation in the cells used here. To this end, the authors cite work by Kulak, 2014 and Ly, 2018 that MCF10A and HELA cells express 10-fold less b3 integrin compared to b1 integrin. While b3 is indeed found at lower levels in these studies it might be (a) nonetheless present at sufficient levels to initiate adhesion formation (as it was picked up by mass-spec analysis); (b) the authors discuss that b1 integrin is 10fold higher expressed compared to Cas ( based on same studies) meaning that Cas and b3 integrin would be expressed at comparable levels, further supporting point (a) above; lastly, (c) the studies by Kulak and Ly find other β-integrins at high levels – Kulak: ITGB5 and ITGB8; Ly: ITGB5, ITGB6, ITGB8, which are all RGD receptors and can complex with ITGAV which is also found at high levels in both studies, consistent with the general understanding of αvβ6 being a major fibronectin receptor in cancer cells. Mechanosensing and binding properties are different for these integrins (e.g. PMID: 24793358, 22328497), but especially the binding to Cas, or their involvement in early adhesion formation have not been characterised to my knowledge. As such the fact that Cas clusters before b1 integrin does not suggest that it is integrin independent, but might only associate with clusters of other integrins (e.g. αvβ6) that then mature into b1 integrin-containing adhesions.

– Apart from this central point, the authors show that phosphorylated Cas is enriched at the adhesome, knockdown of Cas leads to cell spreading and migration defects, that Cas is upstream of vinculin recruitment and that Cas is phosphorylated by Yes1, which are all points that have been described in the past. Especially, inwards movement of Cas dependent on phosphorylation: PMID 24928898; the role of Cas in spreading: PMID 10448062; also here: PMIDs 23974298, 21937722; 15817476;

– The authors find phosphorylation through Yes, but not fyn or src, although PMID 17129785 previously demonstrated that Yes, Src, and Fyn can phosphorylate Cas; whereas PMID 11604500 finds that Src can phosphorylate Cas. The discrepancy here should be explored.

– Also the lack of effect of Src and Fyn knockdown in the current study is in apparent contrast to the existing literature, e.g. PMID 16597701, and should be explored further.

– Similarly the question about the recruitment of vinculin downstream of Cas seems to be in disagreement with the literature reporting that vinculin is upstream of FAK dynamics – i.e. since adhesion localisation of Cas is less pronounced in absence of Vinculin (PMID 23974298).

The authors should expand the investigation to other integrins; in order to exclude that the change in adhesion localisation (i.e. distance from cell edge; e.g. Figure 5C) is not affected by the different penetration depth of the green and red light in the dual color TIRF experiments, the authors should investigate as a control green Cas vs CasmSc.

*Reviewer #3 (Recommendations for the authors):*

The findings of this paper show that epithelial cell adhesion to fibronectin or collagen involves a novel adhesion site assembly process that relies on clusters of Cas at the cell edge. Further, an SFK-Cas-Crk-Rac1 pathway is linked to spreading and further adhesion maturation through vinculin recruitment. The MCF10A and HeLa lines that are used in this study have very low levels of β 3 integrin and the β 1 integrin is the primary integrin involved but Mn activation does not rescue spreading in the absence of Cas.

The mechanism of Cas concentration at the leading is not determined but there are many other findings that support the major hypothesis.

An experiment that would test the hypothesis that the major difference between the adhesion formation in fibroblasts and these epithelial cells is the level of β 3 integrins would be to overexpress β 3 and/or deplete β 1 in the MCF10A cells. If β 3 integrin would be processed and reach the cell surface, then the process of adhesion formation may change.

I liked this paper and it was very thorough in documenting the pathway for adhesion formation. My major suggestion is to try to increase β 3 integrin levels in the epithelial cells or alternatively, deplete them in the fibroblasts while increasing β 1 levels. This may not work for a variety of reasons but it would be spectacular if it did.

---

## [Author Response]

[Editors’ note: the authors resubmitted a revised version of the paper for consideration. What follows is the authors’ response to the first round of review.]

Reviewer #1 (Recommendations for the authors):The authors have investigated the role of the p130-cas/crk complex in cell spreading and the hierarchy of adhesion formation in epithelial cells. The authors propose a model whereby phosphorylated Cas (p130Cas, BCAR1), its binding partner, Crk, and inactive FAK and inactive b1-integrins cluster together at the leading edge of the cell protrusion. This they suggest precedes focal adhesions formation with F-actin, active integrins, active FAK, vinculin, and talin. They show convincingly that p130cas depletion severely impairs cell spreading downstream of integrin activation, as integrin activation with Mn^2+^ fails to rescue the spreading defect. The model is attractive; however, it does not seem to be supported fully by the data. The biggest problem is the unfortunate choice of the integrin antibodies used in the paper. Unfortunately, the authors have been misinformed in their choice of the antibodies used (Please see Byron et al., JCS PMID: 19910492). They indicate that they have chosen an antibody that specifically recognizes the active ligand-bound state of integrins. However, the 9EG7 antibody does not recognize the open conformation but it is specific to the extended primed state. Instead, 12G10 for example would recognize this state.

We thank the reviewer for the supportive comments and apologize for our poor choice of antibodies to characterize integrin activation state. We had mistakenly used 9EG10 to profile active integrin β1, but, as the reviewer points out, 9EG10 actually binds both extended-closed (EC, primed) and extended-open (EO, active) conformations (Su et al. 2016 PMC4941492). We therefore repeated our analysis using 12G10, which is specific for the EO conformation (Su et al. 2016). The 12G10 (EO) and 9EG7 (EO+EC) profiles are very similar, suggesting that levels of EC are low relative to EO (Figure 4D). The results confirm that active (EO) integrin is enriched towards the inner end of each cluster, coinciding with the peak of vinculin, kindlin, talin, and other focal adhesion markers, consistent with our interpretation that this part of the cluster is attached to the ECM.

They also indicate that they have used an antibody that recognizes the bent inactive state that is not ligand bound. Unfortunately, AIIB2 is not conformation-specific but rather blocks directly the ligand binding to the integrins and thus does not stain inactive integrins. This in fact is obvious from looking at the example stainings provided where their inactive beta1 staining is clearly in adhesions. In most epithelial cells, inactive b1-integrin (detected with Mab13, PIH5, or 4B4 antibodies specifically recognizing inactive integrins) localise primarily to membrane ruffles above the FA staining detected by TIRF.

We regret the error. As the reviewer points out, AIIB2 recognizes both active and inactive integrin β1 (Mold et al. 2016, PMC5076510). The AIIB2 profile showed a prominent peak near the outer edge of the cluster, co-localizing with phospho-Cas, and a lower, broad peak overlapping with vinculin. We interpreted these peaks as inactive and active integrin respectively. As the reviewer recommended, we have now used mAb13, which binds both bentclosed (BC) and EC integrin β1 (Su et al. 2016). mAb13 only shows the first peak, co-localizing with phospho-Cas (Figure 4D). Inactive integrin was also detected on the upper cell surface but was out of the plane of focus and was not quantified.

Comparing the profiles obtained with mAb13 (BC+EC), 9EG7 (EO+EC), 12G10 (EO) and AIIB2 (total) suggests that inactive (BC) integrin β1 peaks with Cas and pYCas near the cell edge, while active (EO) β1 peaks with vinculin and other focal adhesion markers further from the edge (Figure 4E). Minimal overlap between mAb13 and 9EG7 profiles implies that the level of primed (EC) integrin β1 is low.

Please note also that the spatial distribution of total integrin β1 (all conformations) across the adhesion is difficult to assess by immunostaining, which is influenced by antibody affinity and accessibility. However, live imaging showed that integrin β1 clustering is delayed relative to Cas (Figure 2), suggesting that immunostaining underestimates the amount of active EO integrin under the cell body relative to inactive BC integrin near the edge.

A big part of the data included in the main figures describes the approaches the authors have used to generate endogenously tagged cell lines for some of the components investigated. This is commendable but as such genomic editing is an established methodology, these details could be moved to the supplements.

We have moved details of cell line construction to the methods.

Throughout the manuscript, the quantifications are based on rather low numbers of cells, and for some experiments, it is not clear if these are from a single experiment or from biologically independent experiments. In addition, there are several instances where the authors show just a small magnified area of the cells. Smaller magnification images should be provided as well to help the reader to evaluate the quality of the staining overall and how representative the chosen areas are.

We have increased cell numbers and added low and high magnification images as needed.

Specific points:1) For endogenous tagging of Cas, the authors employed the insertion of an artificial exon containing mScarlet (mSc) into the first intron of the Cas gene. In the resulting protein, mSc is inserted between Pro4 and Asn5 of Cas without any further linker sequence, thus potentially affecting some of the functions of the Cas SH3 domain.

We apologize for our poor description of the construct. We omitted the important point that the mScarlet PCR product used to generate the targeting construct included 24 nucleotides of vector sequence. This introduces an 8-residue linker (GGMDELYK) between the folded structure of mScarlet and residue 5 of Cas. This makes our construct more similar to the viral constructs used by others. For example, inserting Cas at the BglII site in pEGFP-C1 introduces a 5-residue linker (SGLRS) between EGFP and the first residue of Cas (Janostiak et al. 2011, PMID: 21937722; Branis et al. 2017, PMC 5390273). The reviewer may also be concerned about possible functions of first four residues of Cas (MNHL), which now separated from residue 5 by mScarlet and the linker. In principle, these four residues could be important for Cas function. This seems unlikely because MNHL is just one of seven different alternative first coding exons that all splice into the same second exon (Uniprot P56945). The alternative first exons range in length from 2 to 50 residues and end in various residues – L, W, K, E, P (twice) and R – suggesting that the protein must tolerate a variety of N-terminal sequences. We have modified the Results and Figures to indicate that Cas^mSc^ contains a linker between mScarlet and the SH3 domain.

Even with the linker, mScarlet tagging may affect Cas function. Therefore, we performed additional controls, comparing parental and Cas^mSc^ MCF10A cells. First, we plated parental MCF10A cells on collagen for 30 min and immunostained for phospho-Cas and vinculin. Cell spread area and the areas and intensities of pYCas and vinculin clusters were the same as for Cas^mSc^ MCF10A cells analyzed in parallel (new Figure 1 —figure supplement 2). Second, we analyzed the spatial distribution of endogenous Cas and vinculin in parental MCF10A cells (new Figure 4 —figure supplement 1). The untagged proteins are organized spatially in the same pattern as Cas^mSc^ and EYFP-vinculin (Figure 4). Thus, mScarlet tagging appears not to affect Cas function or spatial organization.

Moreover, phosphorylation of Tyr12 in the Cas SH3 domain has been reported to regulate interaction with several SH3-binding proteins, including vinculin (PMID: 21937722, PMID: 28808245, PMID: 23974298). The authors should provide evidence that inserting mSc in close proximity to the Cas SH3 domain and an important regulation site (Y12) does not affect downstream signalling of the endogenously tagged Cas. For instance, the authors should analyse the interaction of Cas and CasmSc with SH3 domain-binding proteins or analyse Y12 phosphorylation levels of Cas and CasmSc in cells with and without Src expression.

pY12 was identified in a shotgun pTyr proteomics experiment on SrcYF-transformed mouse fibroblasts (Luo et al. 2008, PMC2579752). The investigators then developed a pY12 antibody and found that the site is phosphorylated in some cancer cells and in SrcYF-transformed but not control fibroblasts (Janostiak et al. 2011, PMID: 21937722). However, pY12 has not been noted by other investigators. It has only been detected in three high-throughput phosphoproteomics experiments, compared with 10’s to 100’s of detections of other Cas pY sites (Phosphosite). We suspect that Cas Y12 may only be phosphorylated when Src is highly active. We contacted Dr. Rosel requesting the pY12 Cas antibody used in PMID: 21937722, 28808245 and 23974298.

However, Dr. Rosel declined to send the antibody, stating that it is not very specific and partially recognizes Y12F Cas. There is no commercial source for a pY12 Cas antibody and it would be difficult for us to perform the analysis.

Functional studies implicating Y12 in signaling made use of a vector expressing wildtype or mutant GFP-mCas (mouse Cas) (Janostiak et al. 2011, PMID: 21937722; Janostiak et al. 2014, PMID: 23974298; Gemperle et al. 2017, PMID: 28808245). The Y12E mutation inhibited localization to focal adhesions and binding to FAK, PTP-PEST and vinculin. We transduced MCF10A cells with a similar retroviral construct (EYFP-mCas) in rescue experiments to control for off-target effects of Cas siRNA (see below, Reviewer 1, point 6). The kinetics of EYFP-mCas relative to mCherry-vinculin (new Figure 5 —figure supplement 2A-C) were very similar to the kinetics of Cas^mSc^ and EYFP-vinculin (Figure 1). Moreover, the spatial distribution of EYFPmCas and mCherry-vinculin in adhesions again showed Cas distal to vinculin, as with Cas^mSc^ and EYFP-vinculin or untagged Cas and vinculin in parental cells. If the mScarlet tagging affected Y12 phosphorylation, we would have expected Cas^mSc^ to behave differently from EYFP-mCas or endogenous Cas. Thus we feel that it is unlikely that the tag affects Y12 phosphorylation.

2) In Figure 1B authors show levels of Cas in the CasmSc cells. While most of the Cas in CasmSc cells is tagged, the phosphorylation pattern as shown by pY410 Ab does not replicate the pattern of the total Cas antibody, suggesting phosphorylation of the tagged Cas is substantially weaker than that of the remaining untagged Cas. Authors should provide further evidence that their tagging strategy does not influence Cas substrate domain phosphorylation and thus signalling downstream of the phosphorylated SD (such as interaction with Crk) when compared to untagged Cas.

Thank you for pointing out the unexpected low stoichiometry of phosphorylation of Cas^mSc^ relative to untagged Cas in Figure 1B. These were not typical blots: there may have been a problem with transfer of the upper band. In most blots the phosphorylation stoichiometries of Cas and Cas^mSc^ are equal. We have replaced the panel with blots from a more representative experiment. This result and other controls mentioned in our response to Point 1 (above) lend confidence that tagging Cas does not affect its phosphorylation or function.

3) In Figure 2C the authors show data points of 11 cells on Fn and 7 cells on collagen (although the legend indicates n=7-13). Measurement of more cells should be provided in order to strengthen this observation, given the median Δt_1/2_ (VCL^-^Cas) ranges from less than 0s to more than 120s.

We increased the number of replicates in Figure 2C (now Figure 6C). There is still no significant difference in Cas-vinculin recruitment kinetics on collagen or fibronectin.

Moreover, based on this data authors suggest that "Cas clusters may specify localization of integrin α5β1 and α2β1 adhesions", however, this is not obvious from the data provided in this section.

Previous Figure 1 showed that Cas clustered in membrane regions that developed into focal adhesions when cells were plated on collagen, with a time delay between Cas clustering and vinculin recruitment of about a minute. Previous Figure 2 reported similar findings when cells were plated on fibronectin, and used blocking antibodies to show the importance of integrin α2β1 for collagen adhesion and α5β1 for fibronectin adhesion. Based on this, we speculated that Cas may specify localization of vinculin clusters. However, Cas siRNA results were not shown until later and did not use fibronectin. In response to this comment and comments from Reviewer 2, we have investigated focal adhesion assembly on different ECMs in more detail. The new results are in Figure 6 and Figure 6 —figure supplement 1. Briefly, we found that (a) Both MCF10A epithelial cells and human foreskin fibroblasts (HFFs) require Cas to spread and assemble focal adhesions on both collagen and fibronectin; (b) Cas clusters develop into vinculin clusters after ~1 min delay on both collagen and fibronectin, implying a similar mechanism; (c) Cas-dependent attachment to collagen and fibronectin is inhibited by integrin β1 knockdown or by α2β1-blocking antibody; (d) Cas-dependent attachment to fibronectin is inhibited by either β1 or αv knockdown or α5β1-blocking antibody. Taken together, these results suggest that Cas is required for adhesion formation through different integrin heterodimers on different ECM and in different cell types.

4) Figure 3. Please show the overall distribution of the tagged integrins in the cell. How well do the tagged integrins overlap with the endogenous ones?

We have added lower magnification time-lapse images of a spreading Cas^mSc^ β1ecto-pH MCF10A cell to Figure 2. The tagged integrin distribution is very similar to the endogenous integrin immunostaining (Figure 4 —figure supplement 2).

Please show b1-integrin blots alongside the GFP in the WB.

Figure 3E (now Figure 2E) showed a GFP Western blot of ITGB1^GFP^ cells. It would be ideal to also probe with β1 antibody. We would expect to see a single band in control cells and two bands in ITGB1^GFP^ cells. Unfortunately, after testing two monoclonals and two rabbit antibodies, the only antibody that worked on a Western (AB1952P, Millipore) did not detect the β1-GFP fusion protein. Instead, the amount of untagged integrin β1 was decreased in ITGB1^GFP^ cells (see Author response image 1, black arrowhead). This suggests that one allele has been tagged and is not detected by this antibody. AB1952P was raised to the C terminus of integrin β1, where the tag is attached. Rather than showing this Western, we feel that showing the cDNA sequence, the GFP blot, and GFP in focal adhesions is sufficient evidence that the gene is tagged correctly.

**Author response image 1. sa2fig1:** 

5) Figure 5. The authors need to provide representative images (lower and higher mag) of all the antibody stainings they have quantified in 5C. How has the specificity of the antibodies/staining been determined? Please see above the comments related to the integrin antibodies used.

We have added low and high magnification images for all immunostaining quantified in Figure 4C. Please see new supplements to Figure 4 —figure supplement 1 (Cas and vinculin), figure supplement 2 (integrin β1 conformation epitopes) and figure supplement 3 (CrkL, paxillin, talin1, kindlin2, Yes1, FAK, pY397FAK and pY861FAK). Cas and vinculin antibodies were validated by immunofluorescence and Western blotting of corresponding knockdown cells (Figure 5A and B and Figure 5 —figure supplement 1A). The CrkL, paxillin, talin1, kindlin2, Yes1 and FAK antibodies were validated by Western blot of cells treated with the corresponding siRNAs (Figure 5 —figure supplements 4 and 5).

6) Figure 6 and 7. How have the authors controlled for off-target effects of their siRNAs?

We have not controlled for off-target effects of siRNAs against FAK, Kindlin2 or Talin1 because they had little effect on Cas clustering and their effect on vinculin recruitment agrees with published studies. Regarding SFK siRNAs, we note that the pan-SFK allosteric inhibitor eCF506 inhibited vinculin recruitment to Cas clusters, suggesting that one or more SFKs is required. When we knocked down Src, Fyn and Yes we surprisingly found that only Yes was required, despite good knock down in all cases. We discuss the possibility that Yes has a special role based on its trafficking or preference for membrane microdomains. We did not control for off target effects of Yes siRNA but in view of the eCF506 result it seems likely that the effect of Yes siRNA is on-target.

The vinculin silencing efficacy in Figure 6 is poor and precludes making any conclusions on the role of vinculin.

We agree that vinculin silencing is poor despite use of a double knockdown protocol (Figure 5 —figure supplement 1A). Vinculin may have a longer half life than the other proteins we studied. However, imaging shows that vinculin siRNA caused a ~50% decrease in vinculin mean intensity but no decrease in Cas intensity, while Cas siRNA caused a ~50% decrease in Cas intensity and a ~60% decrease in vinculin intensity. Thus Cas appears to be quantitatively limiting for vinculin clustering but vinculin is not limiting for Cas clustering.

In Figure 6C the authors analyse the role of Cas depletion on the regulation of Mn^2+^-dependent spreading. As a further control, the authors could include a rescue experiment with Cas WT and Cas lacking the CCH domain in Cas-depleted cells.

To control for off-target effects of Cas siRNA, we have now generated cells expressing EYFPmCas (mouse Cas) and mCherry-vinculin, treated with control or human Cas siRNA, and measured cell spread area and the number, areas and intensities of Cas-vinculin clusters in absence and presence of Mn^2+^. Wildtype EYFP-mCas fully rescued all effects of Cas siRNA (Figure 5 —figure supplements 2D and 5). We did not try a CCH domain mutant, but a 15F mutant, lacking the 15 YxxP phosphorylation sites, failed to rescue focal adhesion formation in the absence or presence of Mn^2+^. Clustering of the 15F mutant was also defective, consistent with the importance of Cas phosphorylation for clustering. These results control for off-target effects of siCas and for possible issues of use of red and green fluorophores on detection by TIRF, as well as showing the importance of Cas phosphorylation.

7) The results in Figure 7A suggest that the interaction of Crk with phosphorylated Cas accelerates the recruitment of Vinculin to the Cas-Crk clusters. Do the authors have a mechanistic explanation of how this could be regulated?

Our two-step model is presented in Figure 8E and the Discussion. We propose that SFK-CasCrk/CrkL complexes first activate a Rac1 positive-feedback loop to build cluster size and incorporate inactive integrin. Cluster growth may be driven by the formation of protein networks between multiply phosphorylated Cas bridged by Crk/CrkL to proteins containing multiple SH3 binding sites, such as DOCK180. These protein networks interact with integrins through an unknown mechanism, perhaps bridged by membrane microdomains or Yes1-integrin binding. In a second step, the integrin is activated and structural components such as talin and vinculin are recruited. However, it is unclear how the transition from step 1 to step 2 is regulated. There may be lateral interactions between integrin heterodimers, membrane microdomains and other components that lead to adhesion maturation; Cas may need to be inactivated; or some kind of critical size may need to be reached for force application.

Also, to support this data, the authors should perform a Crk rescue experiment in Crk-depleted cells to see whether recruitment of Vinculin to Cas clusters will be rescued to control levels.

We have not performed the suggested rescue experiment but we have added results of single and double knockdowns of Crk and CrkL (Figure 5 —figure supplement 3A-C). Neither single knockdown had much effect but the double knockdown reduced the number, areas and intensities of Cas-vinculin clusters. This suggests functional redundancy between Crk and CrkL. If there the Crk and CrkL siRNAs have off-target effects, they would also have to be functionally redundant, which seems improbable.

8) In Figure 8D, the fluorescence intensity of CasmSc seems to be lower in siSOCS6 than in siCtrl cells. This is rather counterintuitive since the depletion of SOCS6 should stabilize Cas levels. Could the authors comment on this?

Figure 8D shows that either siSOCS6 or the Neddylation inhibitor MLN4924 shortens the time interval between Cas arrival and vinculin arrival. However, as the reviewer notes, the fluorescence intensity of Cas^mSc^ is decreased, not increased as might be expected if Cas was less degraded. We have performed immunofluorescence and find that pY410Cas is unchanged (i.e., the pYCas/Cas ratio is increased). This is consistent with SOCS6 targeting pYCas for degradation. However, why is total Cas^mSc^ decreased? We note that vinculin intensity also decreased. We know that siSOCS6 stimulates Cas-dependent adhesion *disassembly* as well as assembly (Teckchandani et al. 2016 PMC5092051). Thus we suspect that the decreased intensities of Cas and vinculin at the 30 min time point may be due to quicker turnover.

To explore this possibility further, we have re-analyzed the time-lapse imaging of Cas^mSc^ YFP-VCL cells treated with SOCS6 siRNA (see Author response image 2). The time delay between Cas *departure* and vinculin *departure* is decreased when SOCS6 is depleted. Thus, faster turnover of adhesions in SOCS6-depleted cells may mean that more adhesions are undergoing disassembly when Cas and vinculin intensities were measured at the 30 min time point. We feel that fully understanding this aspect would take more analysis and prefer to leave it for follow up study.

Other points:1) In the introduction, the authors mention the mechanoresponsive roles of talin in adhesion maturation and assembly. It would be beneficial to mention in this context also the effect of the force-mediated extension of the Cas substrate domain on its downstream signalling (PMID: 17129785, reviewed in PMID: 25062607).

The role of force in Cas substrate domain phosphorylation is unclear. Unlike talin, very little energy is needed to extend the Cas substrate domain and it is unlikely to be structured in cells (Hotta et al. 2014, PMID: 24722239). However, allosteric regulation may be important and we have added the Sawada paper to the Introduction.

2) Figure 1 supplement 2. What is the difference between Hela panels E and F?

Now Figure 1 —figure supplement 3, panel E shows the half-maximum Cas-vinculin time delays for 37 clusters in a single spreading HeLa cell. The data are not normally distributed, so we calculate a median time delay (66.2 s in this cell) and variance (95% CI 21-114 s). Panel F shows the median time delays clusters in each of n=13 cells, from which mean time interval (48.08 s) and variance (SEM + 9s) can be calculated. Similar comparison of single and multiple cell data are shown for spreading and migrating MCF10A cells in Figure 1 G and H and Figure 1 —figure supplement 3C and D.

3) It is not clear, whether the cells measured in Figure 3 are seeded on Fn or Col-I. If on Fn – would the delay in ITGB1 arrival to Cas clusters be different on Col-I, since authors show there are more 9EG7-positive ITGB1 clusters in cells seeded on collagen (Figure 2B)?

All experiments in the initial submission except Figure 2 were done on collagen. We’ve expanded our studies of adhesion assembly on different ECM and moved the relevant section later in the paper. The new Figure 6 shows that both epithelial and fibroblast cells require Cas to assemble adhesions on fibronectin and collagen. Cas clusters are precursors for vinculin clusters after a similar time delay on both matrix proteins. Attachment to collagen is inhibited by β1 siRNA or α2β1-blocking antibody, but not by αv siRNA. Attachment to fibronectin is inhibited by αv or β1 siRNA or by α5β1-blocking antibody. Thus, the role of Cas in focal adhesion assembly extends to different cell types, integrins, and matrix proteins. Please also see Reviewer 2, point 1.

4) This study is in line with previous work (PMID: 30639111), where pY410 Cas has been shown to co-localize with Crk in filopodia tips in a CCH domain-dependent manner and extends these observations to the edge of spreading/migrating MCF10A and HeLa cells. This could perhaps be discussed.

In separate experiments, we are analyzing Cas mechanism in regulating filopodia dynamics induced by cell attachment. Since this work is unfinished, we would prefer to leave comparisons between adhesions and filopodia to a future publication.

Reviewer #2 (Recommendations for the authors):The manuscript "Cas phosphorylation regulates focal adhesion assembly" by Kumar et al. investigates the role of Cas in cell spreading and finds that Cas, Crk, and FAK cluster together at the cell periphery to initiate focal adhesion formation. In contrast to the previous understanding, they suggest that Cas clustering happens before integrin clustering and hence is the initial step in adhesion formation. This notion is the key novel aspect of the study, however, it is based on the assumption that b1 integrin is the only relevant integrin in the system, which at this point is not supported by the data.

The reviewer’s main concern is that a different integrin may be present and induce Cas phosphorylation before integrin β1 is activated. However, most of our experiments were done on collagen, and, to the best of our knowledge, all collagen receptors contain β1 with various α chains. Adhesion assembly on collagen was inhibited by blocking antibody to α2β1. Therefore, we did not explore possible roles of other integrins. However, we realize that other integrins may be important during attachment to other ECM. In the revision, we have expanded our study to investigate the roles of other integrins and other ECM. We note that all integrins expressed by MCF10A cells form heterodimers with either β1 or αv, so we have used siRNA to inhibit expression of these integrins and studies cell spreading and adhesion formation. We found that collagen adhesion was inhibited by β1 but not by αv siRNA. Therefore it seems unlikely that a different integrin induces Cas phosphorylation before β1 is activated on collagen. In addition, we found that adhesion to fibronectin requires Cas, αv and β1. We have not tagged αv to follow its clustering kinetics, but immunofluorescence of spreading cells revealed β1 adhesions nearer the edge and adhesions containing both β1 and αv further back, suggesting that αv comes later. Therefore, the evidence points to Cas-dependent collagen attachment through integrins containing β1, and Cas-dependent fibronectin attachment through dimers containing αv and β1, with αv unlikely to precede β1.

Specific comments:– The central finding of the study (i.e. that Cas clusters before integrin) is based solely on the study of b1-integrin and hence on the assumptions that it is the only integrin that is relevant for adhesion formation in the cells used here. To this end, the authors cite work by Kulak, 2014 and Ly, 2018 that MCF10A and HELA cells express 10-fold less b3 integrin compared to b1 integrin. While b3 is indeed found at lower levels in these studies it might be a) nonetheless present at sufficient levels to initiate adhesion formation (as it was picked up by mass-spec analysis); (b) the authors discuss that b1 integrin is 10fold higher expressed compared to Cas ( based on same studies) meaning that Cas and b3 integrin would be expressed at comparable levels, further supporting point (a) above; lastly, (c) the studies by Kulak and Ly find other β-integrins at high levels – Kulak: ITGB5 and ITGB8; Ly: ITGB5, ITGB6, ITGB8, which are all RGD receptors and can complex with ITGAV which is also found at high levels in both studies, consistent with the general understanding of αvβ6 being a major fibronectin receptor in cancer cells. Mechanosensing and binding properties are different for these integrins (e.g. PMID: 24793358, 22328497), but especially the binding to Cas, or their involvement in early adhesion formation have not been characterised to my knowledge. As such the fact that Cas clusters before b1 integrin does not suggest that it is integrin independent, but might only associate with clusters of other integrins (e.g. αvβ6) that then mature into b1 integrin-containing adhesions.

We focused on Cas role in integrin β1 adhesion assembly because most of our experiments used cells spreading or migrating on collagen. To our knowledge, all collagen-binding integrins contain integrin β1. We confirmed β1 dependence using blocking antibody P1E6 to α2β1, which inhibited MCF10A cell adhesion to collagen (previous Figure 2A, now Figure 6 —figure supplement 2A). To further confirm the importance of integrin β1 for collagen adhesion, we have now knocked down β1 in epithelial and fibroblast cells and found that cell spreading and formation of Cas-vinculin clusters was inhibited (new Figure 6 —figure supplement 1B and C). Since other integrins are not known to bind collagen, we think it is unlikely that Cas clusters with other integrins during adhesion assembly on collagen.

We also investigated Cas role in fibronectin adhesion, where αvβ3/β6 and α5β1 may both be important. We previously reported that Cas clustering preceded vinculin clustering by a similar time delay on fibronectin as on collagen, and that attachment and spreading on fibronectin were inhibited by blocking antibody P8D4 to α5β1 (previous Figure 2, now Figure 6B and Figure 6 – supplement 2A). We now show that both epithelial and fibroblast cells require Cas, β1 and αv for attachment and spreading on fibronectin (Figure 6 – supplement 1B and C). This implies that Cas regulates assembly of adhesions containing β1 or αv (or both) on fibronectin.

Immunofluorescence of spreading cells did not reveal any αv clusters that do not also contain β1. Early β1 clusters at the edge lack αv. Therefore, it seems unlikely that an αv integrin induces Cas clustering before β1 recruitment.

– Apart from this central point, the authors show that phosphorylated Cas is enriched at the adhesome, knockdown of Cas leads to cell spreading and migration defects, that Cas is upstream of vinculin recruitment and that Cas is phosphorylated by Yes1, which are all points that have been described in the past. Especially, inwards movement of Cas dependent on phosphorylation: PMID 24928898; the role of Cas in spreading: PMID 10448062; also here: PMIDs 23974298, 21937722; 15817476;

We acknowledge that there have been many previous studies of Cas in focal adhesions but our results shed new light on the role of Cas in adhesion assembly, which has been controversial. Previous adhesome studies were not designed to measure the proportion of a specific protein or phosphoprotein in the adhesome, just whether they were detectable. Our semi-quantitative Western blot analysis shows that pY410Cas, pY249Cas and pY861FAK are highly enriched in the adhesome fraction relative to their non-phosphorylated (or in the case of FAK, pY397 phosphorylated) forms (Figure 4 —figure supplement 4). This suggests that these phosphorylations occur locally within adhesions and are rapidly lost when Cas or FAK dissociate.

Other studies have shown that Cas is required for cell spreading and migration, but we are not aware of other studies investigating how Cas regulates adhesion assembly.

Machiyama et al. (2014) (PMID 24928898) reported that Cas molecules within adhesions sites move inwards dependent on actomyosin contraction and Cas phosphorylation and that Src-Cas dissociation is important for adhesion disassembly. They inferred that Cas may become more phosphorylated towards the inner end of an adhesion, the inverse of what we see. Honda et al. (1999) (PMID 10448062) first reported that cas-/- MEFs spread and assembled the actin cytoskeleton slowly. Janostiak et al. (2014) (PMID 23974298) and 2011 (PMID 2193772) detected direct Cas-vinculin interaction and concluded that vinculin regulates Cas localization, rather than vice versa, as we report. Sanders and Basson (2005) (PMID 15817476) showed that Cas but not paxillin knockdown inhibited Caco2 cell spreading but did not look at adhesion assembly. The Honda and Janostiak contributions were cited in the previous submission and we have added Sanders and Basson to the new submission.

– The authors find phosphorylation through Yes, but not fyn or src, although PMID 17129785 previously demonstrated that Yes, Src, and Fyn can phosphorylate Cas; whereas PMID 11604500 finds that Src can phosphorylate Cas. The discrepancy here should be explored.– Also the lack of effect of Src and Fyn knockdown in the current study is in apparent contrast to the existing literature, e.g. PMID 16597701, and should be explored further.

We agree that a special role for Yes was unexpected. Sawada et al. (2006) (PMID 17129785) stated (but did not show) that Cas phosphorylation in SYF (src-/- fyn-/- yes-/-) MEFs was restored by expressing cSrc, cYes or Fyn and showed that recombinant cSrc or FynT could phosphorylate Cas in vitro. Ruest et al. (2001) (PMID 11604500) is one of several publications reporting that SFKs phosphorylate Cas with FAK as a scaffold. Kostic and Sheetz (2006) (PMID 16597701) showed Fyn dependent phosphorylation of Cas in αvβ3 adhesions forming at the edge of fibroblasts spreading on fibronectin. We have added Discussion of how differences in affinity for integrin tails or for lipid microdomains may explain the unexpected role for Yes.

– Similarly the question about the recruitment of vinculin downstream of Cas seems to be in disagreement with the literature reporting that vinculin is upstream of FAK dynamics – i.e. since adhesion localisation of Cas is less pronounced in absence of Vinculin (PMID 23974298).

As stated above, Janostiak et al. (2014) (PMID 23974298) concluded that vinculin regulates Cas localization, rather than vice versa, as we report. They measured the % of paxillin-positive focal adhesions that contained GFP in GFP-Cas-expressing control and vinculin-/- MEFs. The focal adhesions in vinculin-/- MEFs were smaller than in control MEFs which may have made it harder to detect co-localization of GFP-Cas against the cytoplasmic background. Given the importance of vinculin in reinforcing talin-actin connections, vinculin-/- MEFs may have upregulated compensatory pathways during selection for growth in culture.

The authors should expand the investigation to other integrins.

Please see response to comment 1 above for discussion of other integrins.

In order to exclude that the change in adhesion localisation (i.e. distance from cell edge; e.g. Figure 5C) is not affected by the different penetration depth of the green and red light in the dual color TIRF experiments, the authors should investigate as a control green Cas vs CasmSc.

Regarding choice of red and green fluorophores: we obtain similar results from studies of EYFPmCas and mCherry-VCL (Figure 5 —figure supplement 2).

Reviewer #3 (Recommendations for the authors):The findings of this paper show that epithelial cell adhesion to fibronectin or collagen involves a novel adhesion site assembly process that relies on clusters of Cas at the cell edge. Further, an SFK-Cas-Crk-Rac1 pathway is linked to spreading and further adhesion maturation through vinculin recruitment. The MCF10A and HeLa lines that are used in this study have very low levels of β 3 integrin and the β 1 integrin is the primary integrin involved but Mn activation does not rescue spreading in the absence of Cas.The mechanism of Cas concentration at the leading is not determined but there are many other findings that support the major hypothesis.An experiment that would test the hypothesis that the major difference between the adhesion formation in fibroblasts and these epithelial cells is the level of β 3 integrins would be to overexpress β 3 and/or deplete β 1 in the MCF10A cells. If β 3 integrin would be processed and reach the cell surface, then the process of adhesion formation may change.I liked this paper and it was very thorough in documenting the pathway for adhesion formation. My major suggestion is to try to increase β 3 integrin levels in the epithelial cells or alternatively, deplete them in the fibroblasts while increasing β 1 levels. This may not work for a variety of reasons but it would be spectacular if it did.

We thank the reviewer for their kind comments.

Many previous studies of focal adhesion assembly mechanism and kinetics have made use of fibroblasts attaching to fibronectin through αvβ3 as well as α5β1. Formation of tension-sensitive, αvβ3 adhesions on fibronectin requires Fyn and Cas (von Wichert et al., 2003; Kostic and Sheetz, 2006). However, our experiments were done on collagen and β3 integrins are not known to mediate adhesion to collagen. Moreover, our new siRNA experiments show that β1 but not αv is needed for Cas-dependent epithelial or fibroblast cell attachment to collagen. Together with our finding that α2β1 antibody blocks adhesion, the results suggest that Cas is working through β1 without participation by other integrins on collagen.

The situation is more complicated on fibronectin. We find that β1 siRNA or α5β1-blocking antibody inhibits Cas-dependent adhesion of epithelial cells or fibroblasts to fibronectin. However, siRNA against αv is also inhibitory, suggesting a requirement for αvβ3/6 as well as α5β1. The αv partner may be β6 in epithelial cells since β3 was not detected. The αv partner in fibroblasts may be β3, since we detected β3, αv and β1 in fibroblast focal adhesions. Thus Cas may regulate focal adhesion assembly through integrin dimers containing αv as well as β1. We have not tagged αv so do not know whether it clusters before or after Cas, but we suspect it clusters after Cas and after β1 because immunofluorescence of spreading cells shows αv adhesions further from the cell edge than β1 adhesions.

We are presently investigating how phospho-Cas/Crk clusters nucleate integrin clusters, focusing on β1. For the present paper, we hope the reviewers will agree that showing the Cas requirement for focal adhesion formation on different ECMs, through different integrins, and in two cell types, is sufficiently interesting for publication in *eLife*.